# HuR controls glutaminase RNA metabolism

Douglas Adamoski [1,2], Larissa M. dos Reis [1,2,3], Ana Carolina Paschoalini Mafra [1,2,7], Felipe Corrêa-da-Silva[2,3], Pedro Manoel Mendes de Moraes-Vieira [3], Ioana Berindan-Neagoe [4], George A. Calin [5,6] & Sandra Martha Gomes Dias [1] ✉

Glutaminase (GLS) is directly related to cell growth and tumor progression, making it a target for cancer treatment. The RNA-binding protein HuR (encoded by the *ELAVL1* gene) influences mRNA stability and alternative splicing. Overexpression of *ELAVL1* is common in several cancers, including breast cancer. Here we show that HuR regulates GLS mRNA alternative splicing and isoform translation/stability in breast cancer. Elevated *ELAVL1* expression correlates with high levels of the glutaminase isoforms C (GAC) and kidney-type (KGA), which are associated with poor patient prognosis. Knocking down *ELAVL1* reduces KGA and increases GAC levels, enhances glutamine anaplerosis into the TCA cycle, and drives cells towards glutamine dependence. Furthermore, we show that combining chemical inhibition of GLS with *ELAVL1* silencing synergistically decreases breast cancer cell growth and invasion. These findings suggest that dual inhibition of GLS and HuR offers a therapeutic strategy for breast cancer treatment.

Glutamine is a key nutrient for cancer cells, providing carbons for lipid, amino acid, and ATP synthesis through the TCA cycle. Moreover, glutamine metabolism provides nitrogen for nucleotide's nitrogenous base and amino acid synthesis and participates in glutathione and NADPH production, which is relevant for maintaining redox balance[1]. Glutaminase converts glutamine into glutamate, which is further converted to α-ketoglutarate, a key metabolite for replenishing TCA cycle intermediates, especially in cells undergoing aerobic glycolysis (known as the Warburg effect)[2]. Glutaminase is a promising target for treating many tumor types. One glutaminase inhibitor, telaglenastat (CB-839)[3–6], is under phase I/II clinical trials[6] for multiple solid and hematopoietic tumors, including breast cancers.

Glutaminase is encoded by the *GLS* and *GLS2* genes. *GLS2* forms two isoforms, liver-type glutaminase (LGA) and glutaminase B (GAB), and is conditionally pro-tumorigenic[7]. *GLS* undergoes alternative splicing to produce two transcripts: KGA and GAC. The *GLS* gene presents 19 exons; both GAC and KGA possess exons 1-14. The insertion of exon 15 generates the GAC isoform, while the skipping of exon 15, followed by the inclusion of exons 16–19, generates KGA[3,8]. Both isoforms present a conserved glutaminase domain similar to that of bacterial glutaminases[9], which is flanked by a long N-terminal domain folded in an EF-hand-like four-helix bundle and a C-terminal domain[9]. KGA presents a long C-terminus with three putative ankyrin repeats, whereas GAC possesses a short unstructured C-terminus. Moreover, KGA and GAC also carry distinct 3′ and 5′ untranslated regions (UTRs)[10]. The GAC isoform, compared to KGA, is more highly expressed in many tumor types[11,12], which may be related to the fact that GAC is more catalytically active[12–14].

*GLS* is regulated transcriptionally, posttranscriptionally, and posttranslationally through various mechanisms[8,15–17]. C-Jun, for example, is a direct transcriptional regulator of *GLS* expression[15]. On the other hand, the oncogenic transcription factor c-Myc

[1]Brazilian Biosciences National Laboratory (LNBio), Brazilian Center for Research in Energy and Materials (CNPEM), Campinas, Sao Paulo, Brazil. [2]Graduate Program in Genetics and Molecular Biology, Institute of Biology University of Campinas (UNICAMP), Campinas, Sao Paulo, Brazil. [3]Department of Genetics, Evolution, Microbiology, and Immunology, Laboratory of Immunometabolism, Institute of Biology, University of Campinas-UNICAMP, Campinas, SP, Brazil. [4]Research Center for Functional Genomics, Biomedicine and Translational Medicine, University of Medicine and Pharmacy "Iuliu-Hatieganu", Cluj-Napoca, Romania. [5]Department of Experimental Therapeutics, The University of Texas MD Anderson Cancer Center, Houston, TX, USA. [6]Center for RNA Inference and Non-Coding RNAs, The University of Texas MD Anderson Cancer Center, Houston, TX, USA. [7]Present address: Department of Radiation Oncology, Washington University School of Medicine, S. Louis, MO, USA. ✉e-mail: sandra.dias@lnbio.cnpem.br

transcriptionally represses miR-23a and miR-23b, resulting in higher expression of the GAC isoform[18]. In pancreatic cancer, the long non-coding RNA (lncRNA) GLS-AS, an antisense RNA priming intron 17 of the *GLS* gene, decreases mRNA stability and protein levels[19]. Colon cancer transcript 2 (CCAT2), a lncRNA commonly overexpressed in colorectal cancer (CRC), regulates the alternative splicing of *GLS*, favoring GAC expression over KGA[8]. CFIm25 binding to a poly(A) site within intron 14 favors GAC splicing through an alternative mechanism[20]. Other ncRNAs and regulatory proteins are expected to regulate *GLS* splicing.

The neuron-exclusive[21] RNA binding protein (RBP) HuD (coded by the gene *ELAVL4*) binds to *GLS* intron 14 and regulates its splicing, favoring GAC formation[17]. HuR (coded by the gene *ELAVL1*), on the other hand, is a more ubiquitous paralogue[22]. *GLS* mRNA levels have been shown to respond to acidosis[23], a mechanism mediated by its 3′-UTR region[23–25]. Although not demonstrated, HuR has been proposed to be implicated in glutaminase mRNA stabilization during metabolic acidosis[26] through interactions with AU-rich elements (AREs). Transcriptomic analysis has revealed HuR as a widespread RNA regulator that binds not only to the 3′-UTR[27] but also to the intronic ARE regions of several genes across the human genome[27,28]. As HuR binds to introns, it has been proposed as a splicing regulator. Several cancer-related transcripts containing ARE sequences at their UTR regions, such as proto-oncogenes, cytokines, growth factors, and invasion factors, have been characterized as HuR targets[29–32]. HuR has been proposed as a drug target for multiple cancers[22,33], usually in a combination therapy approach[34–36]. Since glutaminase isoform choice is an essential event in physiological and disease states and HuR is known as a bonafide RNA metabolism regulator, we hypothesized that HuR regulates *GLS* RNA metabolism in cancer.

In this work, using The Cancer Genome Atlas (TCGA) database, we confirm that *ELAVL1* is overexpressed in many cancers, including breast cancer. In breast cancer, increased *ELAVL1* expression relates to a poor prognosis in patients. By using publicly available RNA-seq and RNA immunoprecipitation (RIP)-seq data, we predict the role of HuR in GLS splicing and confirm its binding to its mature mRNA. We manipulate *ELAVL1* expression levels in breast cancer cell lines and show that higher HuR levels correlate with increased KGA levels. We further show that HuR binds to GLS mRNA and use reporter-based assays to confirm HuR is a splicing factor of GLS and a regulator of mRNA stability. As a consequence, knocking down *ELAVL1* increases GAC levels and glutamine addiction. Finally, we show that combining *ELAVL1* silencing with glutaminase inhibition further impairs breast cancer cell growth, migration, and invasion. We propose that double targeting of glutaminase and HuR may have a therapeutic benefit for treating breast cancer.

## Results

### *ELAVL1* is overexpressed in multiple tumors and correlates with poor prognosis and GAC and KGA mRNA levels in breast cancer patients

We compared the expression of *ELAVL1* in 21 tumor studies with paired normal data available from the TCGA databank. We found that *ELAVL1* was significantly overexpressed in 18 tumors (Kolgomorov-Smirnov FDR < 0.05) (Fig. 1a and Supplementary Data 1), including breast cancer (BRCA), compared to normal controls; this difference was sustained in 16 tumors, including breast cancer (among others), when a paired normal: tumor comparison was made (Supplementary Fig. 1a and Supplementary Data 1). In BRCA, following a biomarker cutoff optimization method[37] (Supplementary Fig. 1b), we observed a significantly poorer prognosis (evaluated as progression-free interval[38]) in patients bearing tumors with high *ELAVL1* expression (Fig. 1b) than in patients bearing tumors with low *ELAVL1* expression, corroborating previous studies using independent cohorts[39–43]. Of note, since *ELAVL1*, among the other members of *ELAVL* family is the most expressed one

in BRCA tissues (Supplementary Fig. 1c) and has been implicated in breast cancer[44–46], we decided to explore the relationship between HuR and metabolism. We first reduced the dimensionality of a set of 29,072 gene-set signatures for all BRCA patients from TCGA and verified that higher *ELAVL1* expression was enough to distinguish a UMAP group (Fig. 1c, left). We then performed a spatial clustering using 686 pathways with "metabolic," "metabolism," and similar words, which, again, revealed a distinct, higher *ELAVL1* expression level group (Fig. 1c, right). Correlation analysis employing KEGG metabolic pathways gene-set scores and *ELAVL1* expression levels (Fig. 1d) led to a clustering between *ELAVL1* and pyrimidine, nitrogen, and glyoxylate metabolic scores; Curiously, all these pathways show the metabolization of glutamine as a key feature.

Since HuR is related to glutaminase RNA levels, the key enzyme in glutamine processing, we evaluated whether *ELAVL1* levels were related to GAC and KGA mRNA levels. Using TCGA BRCA data, we verified that tumor samples with high *ELAVL1* expression (Fig. 1f) also presented increased KGA and GAC mRNA levels (Fig. 1g, left and right, respectively); *ELAVL1*-KGA (Pearson's $R = 0.2027$, $p < 0.0001$) and *ELAVL1*-GAC (Pearson's $R = 0.3170$, $p < 0.0001$) mRNA levels were positively correlated with each other (Fig. 1h, left and right, respectively). On the other hand, *ELAVL1* mRNA levels did not positively correlate with *CCAT2*[8] and *GLS2*[7] levels (Supplementary Fig. 1d). Finally, by performing qPCR in a second cohort of 45 nonpaired normal and tumor breast cancer samples, we confirmed that GAC mRNA levels were enhanced in tumors compared to normal tissues (Supplementary Fig. 2, rightmost). The reshaping of overall metabolism in patients with elevated *ELAVL1* expression is expected since increased *ELAVL1* levels is correlated to the upregulation of several genes associated with the glycolysis pathway, TCA cycle, malate-aspartate shuttle, lipid metabolism, among others (Fig. 1e). In conclusion, *ELAVL1* mRNA levels are enhanced in BRCA tumor samples compared to normal tissues, and increased *ELAVL1* levels are related to poor prognosis in patients. Importantly, *ELAVL1* levels are positively correlated with KGA and GAC mRNA levels. Of importance, transcriptomic analysis of breast cancer tissues points to a correlation between HuR and broader metabolic wiring.

### Transcriptomic data show that HuR binds to intron 14 of *GLS*

We evaluated HuR's role as a pre-mRNA splicing regulator by using RNA immunoprecipitation (IP) followed by RNA-seq of bound RNA (RIP-Seq) data (publicly available from the ENCODE project[47]) obtained from GM12878, a lymphoblastoid cell line, and K562, a myelogenous leukemia cell line. To discover HuR's putative splicing regulation regions, we looked for HuR binding sites in introns neighboring alternatively spliced exons (Supplementary Fig. 3a). First, we found that HuR-bound introns present between alternatively spliced exons are spread along the whole genome (Supplementary Fig. 3b). Second, we found a positive correlation between the HuR IP:IgG IP fold-change values (defined as the ratio between the number of reads in HuR IP over IgG IP) for each transcript in both cell lines (Pearson correlation of 0.5472), revealing biological consistency (Supplementary Fig. 3c). However, since the GM12878 data provided a higher number of significantly enriched introns (fold-change > 3 and FDR below 0.05, Supplementary Fig. 3d), we chose the GM12878 dataset for the following analysis.

Next, we used publicly available RNA-Seq data from HeLa cells obtained from control and *ELAVL1* knockdown cells[27] to assess differential exon expression and predict genes regulated by HuR. Strikingly, many genes whose exons were differentially expressed in the *ELAVL1* knockdown samples presented GO biological processes and molecular functions related to RNA metabolism, especially nucleic acid metabolism (Supplementary Fig. 4a). Finally, we compared genes that presented intronic HuR binding sites (detected by RIP-Seq analysis) with those whose exons were differentially expressed when *ELAVL1* was knocked down (detected by RNA-seq data) and defined a list of 175

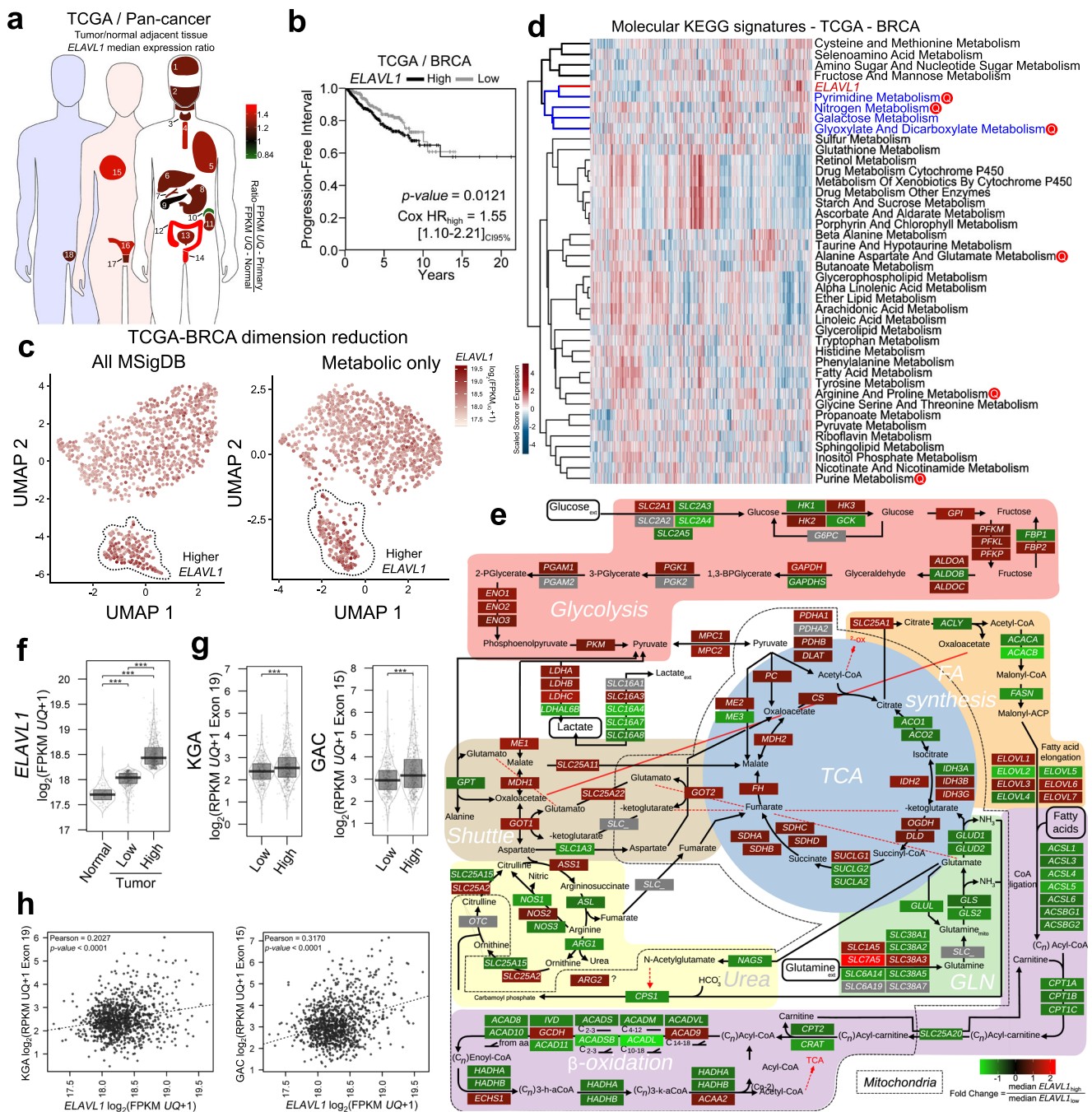

overlapping genes potentially regulated by HuR through alternative splicing. *GLS* was among the genes in this list (Supplementary Fig. 4b and Supplementary Data 2), with HuR presenting a putative binding site for intron 14 (Supplementary Data 3). The influence of HuR levels on metabolism was further assessed by comparing MIA PaCa-2 control cells with *ELAVL1* knockout[48] and resulted in reduced *ELAVL1* (even the non-function allele) and non-significant change on *GLS* overall transcript levels (Fig. 2a). The KEGG metabolic pathways gene signature analysis cleared separated control and knockout samples and clustered *ELAVL1* levels with pathways that involve glutamine metabolism, such as nitrogen and purine metabolism (Fig. 2b). The RNA-seq of HeLa *ELAVL1* knockdown samples revealed that HuR suppression led to an up to 25% decrease in KGA's exclusive exon usage compared to that in the control (Fig. 2c, below), and more than 2.5 fold for MIA PaCa-2 model (Fig. 2c, above). In conclusion, RIP-Seq and RNA-Seq data analysis revealed that HuR affects the *GLS* alternative splicing process by binding to intron 14.

**HuR binds to *GLS* transcripts and influences the isoform levels**
To confirm that HuR impacts *GLS* isoforms, both at the protein and mRNA levels, we knocked down *ELAVL1* in 6 breast cancer cell lines (SKBR3, BT549, MDA-MB-231, MDA-MB-157, HCC38, and Hs578t). Constitutive knockdown of *ELAVL1* (with a sequence called sh*ELAVL1* I) led to substantial depletion of the KGA isoform in two telaglenastat-resistant cell lines, BT549, and MDA-MB-157, which was accompanied by an increase in GAC isoform (Fig. 2d). On the other hand, *ELAVL1* knockdown of the telaglenastat-sensitive cell lines HCC38 and MDA-MB-231[46], which already have a more pronounced GAC level, revealed only a decrease on KGA (Fig. 2d). Then, we used a second shRNA sequence (*ELAVL1* II) and a doxycycline (Dox) inducible system to confirm this result in MDA-MB-231 and BT549 cells, as well as in a third cell line, Hs578t, by western blotting (Fig. 2d and Supplementary Fig. 5a) and immunofluorescence (Supplementary Fig. 5b), and in SKBR3 non-TNBC cell line (Supplementary Fig. 5a). Finally, to provide further proof that HuR controls KGA levels, we used the CRISPR-Cas9

**Fig. 1 | *ELAVL1* expression is increased in multiple tumors and is directly related to GAC and KGA mRNA levels. a** A heatmap of the *ELAVL1* median expression ratio in tumor/adjacent normal tissues throughout the body was produced using TCGA data of 21 tissues. Data from tumors from the same organ were combined. **b** Kaplan-Meier analysis comparing the progression-free interval of TCGA-BRCA patients according to *ELAVL1* expression. High and low *ELAVL1* expression levels were determined using a biomarker cutoff optimization method[37]. **c** Uniform Manifold Approximation and Projection (UMAP) of singscores[111] from each TCGA-BRCA patient using all MSigDB signatures (left) or signatures containing "metabol" as pattern (right), each patient were colored according to *ELAVL1* expression. **d** Heatmap with bidirectional clustering of singscores[111] from KEGG Metabolic Pathways and scaled *ELAVL1* expression. The letter 'Q' marks pathways that involve glutamine. The dendrogram is truncated at the top, and the left side shows the full clustering based on correlation values. **e** TCGA breast cancer tumors were divided into high and low *ELAVL1* expression groups, and the fold change of median values was visualized as a heatmap. Background colors indicate key metabolic pathways such as glycolysis, fatty acid synthesis, TCA cycle, malate-aspartate shuttle, glutamine uptake, urea cycle, and β-oxidation. Dashed line circles indicate mitochondrial biochemical reactions. **f** *ELAVL1* expression is enhanced in breast tumors compared to normal samples. **g** GAC and KGA mRNA levels are increased in breast

tumor samples with high *ELAVL1* levels. For **f** and **g**, 1085 tumors and 113 normal controls from TCGA. **h** *ELAVL1* correlated positively with GAC and KGA mRNA levels in the TCGA-BRCA cohort. Box plots represent the interquartile range; the vertical curve is the kernel density of the distribution, and the dark horizontal line denotes the median. Each dot represents an individual sample. Statistical significance was derived from the log-rank test (B), ANOVA followed by Tukey's test (C), Two-sided Welch's t-test (D), and Pearson correlation test (E). *$p < 0.05$, **$p < 0.01$, ***$p < 0.0001$. Organ numbering on (**a**): 1 - glioblastoma multiforme (GBM), 2 - head and neck squamous cell carcinoma (HNSC), 3 - thyroid carcinoma (THCA), 4 - esophageal carcinoma (ESCA), 5 - pan-lung (lung adenocarcinoma - LUAD and lung squamous cell carcinoma - LUSC), 6 - liver hepatocellular carcinoma (LIHC), 7 - cholangiocarcinoma (CHOL), 8 - stomach adenocarcinoma (STAD), 9 - pancreatic adenocarcinoma (PAAD), 10 - pheochromocytoma and paraganglioma (PCPG), 11 - pan-kidney (kidney renal clear cell carcinoma - KIRC, kidney chromophobe - KICH, and kidney renal papillary cell carcinoma - KIRP), 12 - colon adenocarcinoma (COAD), 13 - bladder urothelial carcinoma (BLCA), 14 - rectum adenocarcinoma (READ), 15 - breast invasive carcinoma (BRCA), 16 - uterine corpus endometrial carcinoma (UCEC), 17 - cervical squamous cell carcinoma and endocervical adenocarcinoma (CESC), 18 - prostate adenocarcinoma (PRAD).

knock-in system to modify the *GLS* gene by introducing a fluorescence protein (mKO2[49]) within exon 19 to express KGA fused to mKO2 at its C-terminus (Fig. 2e). After confirming by western blotting that KGA was successfully tagged (Supplementary Fig. 5e, where the KGA band level was strengthened by synchronizing cells in the S phase[50,51]), fluorescence microscopy showed that *ELAVL1* ectopic expression (Supplementary Fig. 5d, above) increased mKO2 fluorescence (Fig. 2f, above), while *ELAVL1* knockdown (sh*ELAVL1*) (Supplementary Fig. 5d, below) decreased the signal (Fig. 2f, below). As expected, KGA levels reached the highest level on G1-phased cells[50] (Fig. 2g).

To confirm that GAC and KGA protein alterations were related to changes in their mRNA levels, we performed qPCR in BT549 cells. Constitutive knockdown of *ELAVL1* (sh*ELAVL1*) led to a 12.5-fold reduction in *ELAVL1* levels; curiously, KGA mRNA levels presented a 5.3-fold decrease, which was mirrored by a two-fold increase in GAC levels, while *GLS* total mRNA levels did not change significantly (Supplementary Fig. 6a). The impact of Dox-induced *ELAVL1* knockdown using sh*ELAVL1* I and sh*ELAVL1* II on KGA mRNA levels was confirmed in BT549 cells (3.7- and 1.3-fold decrease in KGA mRNA levels compared to those in the control, respectively; Fig. 3a) and MDA-MB-231 cells (1.3- and 2.2-fold decrease in KGA mRNA levels compared to those in the control, respectively; Supplementary Fig. 6b); the effect of sh*ELAVL1* was also confirmed in Hs578t cells (1.5-fold decrease in KGA mRNA level; Supplementary Fig. 6c). *ELAVL1* knockdown significantly increased GAC mRNA levels at variable levels under some conditions.

Having found that *ELAVL1* knockdown led to a decrease in KGA followed by an increase in GAC (sometimes in differing proportions), we went on to confirm HuR binding to *GLS* transcripts. The lack of proportionality in the shifts in GAC/KGA levels led us to speculate that HuR might affect these mRNAs at different steps of RNA metabolism. HuR immunoprecipitation (Fig. 3b and Supplementary Fig. 5h) followed by RT-PCR and qRT-PCR confirmed that HuR binds to different portions of KGA and GAC 3′-UTR sequences in prostate and breast cancer cellular models (Figs. 3c, e and 3d, f, respectively). HuR also binds to the *GLS* pre-mRNA intron 14 in prostate and breast cancer cell lines (Fig. 3g, h, respectively). We then performed fluorescence recovery after photobleaching (FRAP) assay to confirm the in vitro binding of HuR with intron 14 sequence. Recombinant purified HuR-mKO2 bound to in vitro transcribed intron 14, since HuR-mKO2-intron 14 incubation had a half-life of recovery after photobleaching of 25.2 s, slower than what was measured when protein was incubated with control RNA (17.2 s) (Fig. 3i). We concluded that HuR affects both KGA and GAC protein and mRNA levels and binds to GAC and KGA's 3′-UTR and intron 14 of *GLS* pre-mRNA.

## HuR regulates both *GLS* splicing and KGA and GAC mRNA stability

Since HuR binds to *GLS* intron 14 and controls its isoform levels, we speculated that HuR controls *GLS* splicing. To evaluate this hypothesis, we used a splicing reporter system called RG6[8,52], where intron 14 was cloned between two exons. RG6 can produce two fluorescent proteins, DsRed or eGFP, depending on the splicing events in place. In our construct, if intron 14 induces reporter exon 2 retention, the DsRed sequence becomes frameshifted, and only eGFP is correctly coded, indicating GAC formation. On the other hand, if intron 14 promotes reporter exon 2 removal, the DsRed sequence is re-established, producing a stop codon that avoids eGFP translation, indicating KGA formation (Fig. 4a, above). *ELAVL1* inducible knockdown in BT549 and MDA-MB-231 cells with either sh*ELAVL1* I or sh*ELAVL1* II (+Dox) led to a decrease in the DsRed/eGFP fluorescence ratio, indicating exon 2 retention and a preference for GAC formation, compared to that in the control (-Dox, Fig. 4a bellow, on the left and right, respectively). Equivalent results were obtained in Hs578t cells (inducible knockdown with sh*ELAVL1* II; Supplementary Fig. 7a), HEK293T cells (stable knockdown with sh*ELAVL1*, Supplementary Fig. 7b, above; HuR ectopic expression increased the DsRed/GFP signal ratio, Supplementary Fig. 7b, below), BT549 cells (stable knockdown with sh*ELAVL1* and sh*ELAVL1* II; Supplementary Fig. 7c) and SKBR3 cells (stable knockdown with sh*ELAVL1*; Supplementary Fig. 7d).

We then evaluated HuR's effect on GAC and KGA mRNA metabolism. We used a luciferase-based reporter assay in which both KGA and GAC 3′-UTR sequences were cloned after the *Renilla* sp. luciferase gene. Normalization of the transfection efficiency was performed by a firefly luciferase encoded by the same plasmid (Fig. 4b). Following *ELAVL1* inducible knockdown, we verified a 1.5-fold and 1.3-fold decrease in the luciferase signal when the reporter-KGA 3′-UTR and reporter-GAC 3′-UTR constructs were transfected into the cells (Fig. 4b, below, on the left). Consistently, transient ectopic expression of HuR increased the luciferase signal by 1.8-fold when the reporter-KGA 3′-UTR construct was transfected (Fig. 4b, below, on the right). Since HuR can affect both mRNA stabilization and translation[53,54], we sought to study those mechanisms independently. First, we performed mRNA decay experiments following actinomycin D transcription blockade to directly evaluate mRNA stability. Cells transiently expressing HuR provided KGA mRNA, but not GAC mRNA, with greater stability than that in the control condition (empty plasmid) (Fig. 4c). We also silenced *ELAVL1* using a dox-inducible system and observed decreased KGA and GAC's stability (Fig. 4d). To evaluate the HuR effect on translation, we performed a polysome enrichment study (Fig. 4e and Supplementary Fig. 7f). *ELAVL1* silencing demobilized both GAC

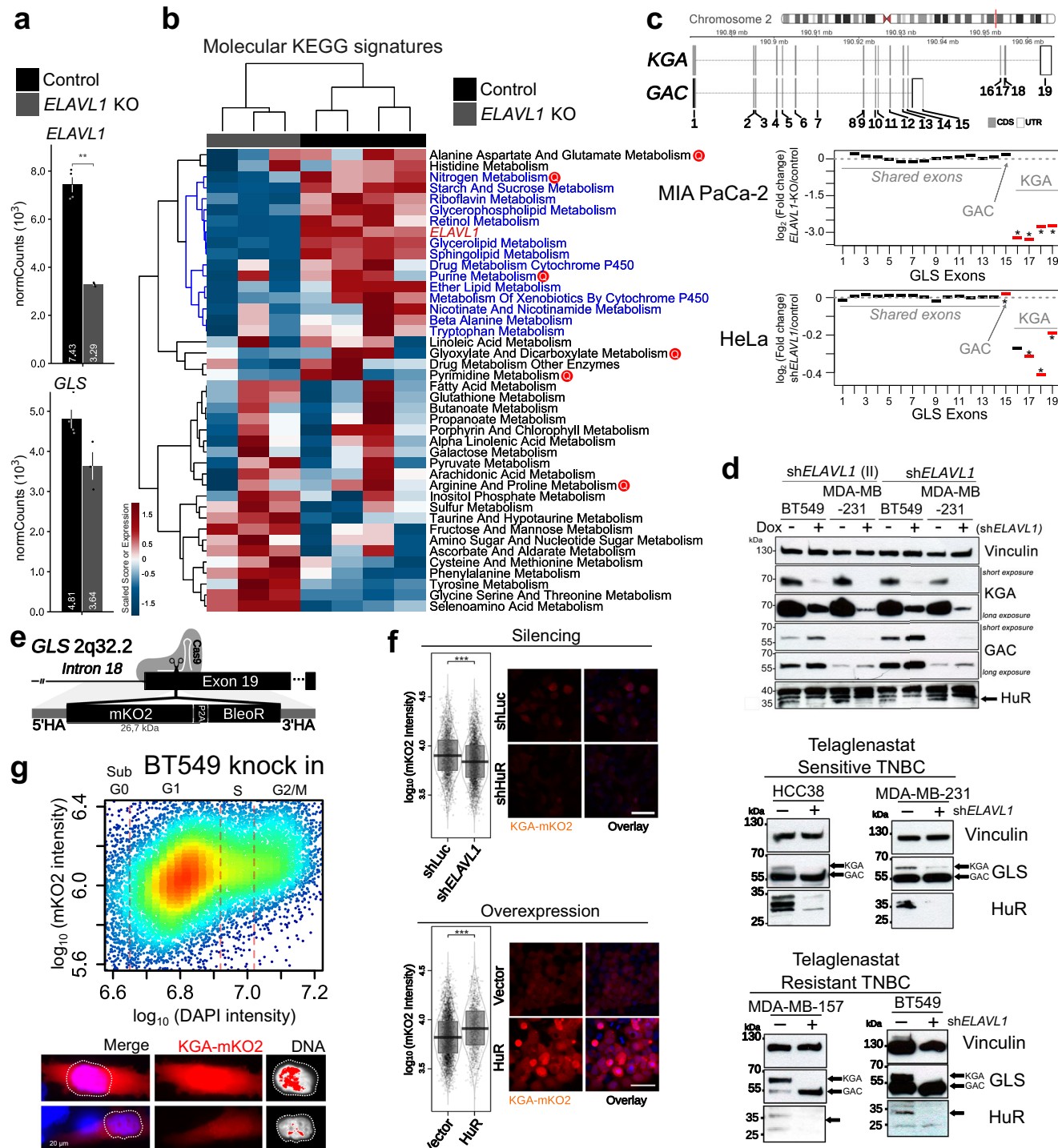

**Fig. 2 | HuR regulates GAC and KGA isoform levels.** MIA PaCa-2 expression levels of *ELAVL1* and *GLS* from RNA-Seq data (**a**, mean and S.E.M.) and heatmap (**b**) with bidirectional clustering of singscores[111] from KEGG Metabolic Pathways and scaled *ELAVL1* expression. The letter 'Q' marks pathways that involve glutamine from (Control *n* = 4; Knockout *n* = 3). The dendrograms represent full bidirectional complete clustering based on correlation values. **c** Chromosome 2 indicating *GLS* loci and exon structure for KGA and GAC isoforms. Bellow, exon differential expression after *ELAVL1* knockdown in MIA PaCa-2 or HeLa cells. Asterisks denote FDR < 0.05. **d**, above, Western blot of doxycycline-inducible *ELAVL1*-silenced breast cancer cell lines (MDA-MB-231 and BT549) using two shRNAs, displaying a marked decrease in KGA protein level, accompanied by an increase in GAC. **d**, below, Western blot of constitutively silenced *ELAVL1* breast cancer cell lines, grouped by telaglenastat resistance, which was defined elsewhere[46]. Arrows indicate KGA and GAC isoforms and HuR-specific bands and western blots repeated at least two times

with reproducible results. **e** Scheme for CRISPR-mediated knock-in of the mKO2-P2A-BleoR cassette into GLS exon 19 to produce the KGA-mKO2 fusion protein. **f** Increase and decrease in cellular fluorescence after *ELAVL1* ectopic expression (bottom) or sh*ELAVL1*-mediated transient silencing (top), respectively, in HEK293T knock-in cells, number independent cells evaluated per condition >1433. Representative images on the right. The scale bar is 50 μm. **g** Cell cycle analysis of BT549 knock-in cells KGA levels (evaluated by mKO2 fluorescence) are enhanced in S-G2/M phases, as expected[50,51]. Representative images in bottom part, with HiLo lookup table for DNA staining, one cell with higher DNA content and higher KGA content and the opposite example below, confirming the functionality of the system. Box plots represent the interquartile range; the vertical curve is the kernel density of the distribution, and the dark horizontal line denotes the median. Each dot represents an individual cell. Statistical significance derived from bootstrapped[105] DEXSeq[107] (**c**) or Two-sided Welch's *t*-test (**f**). *$p < 0.05$, **$p < 0.01$, ***$p < 0.0001$.

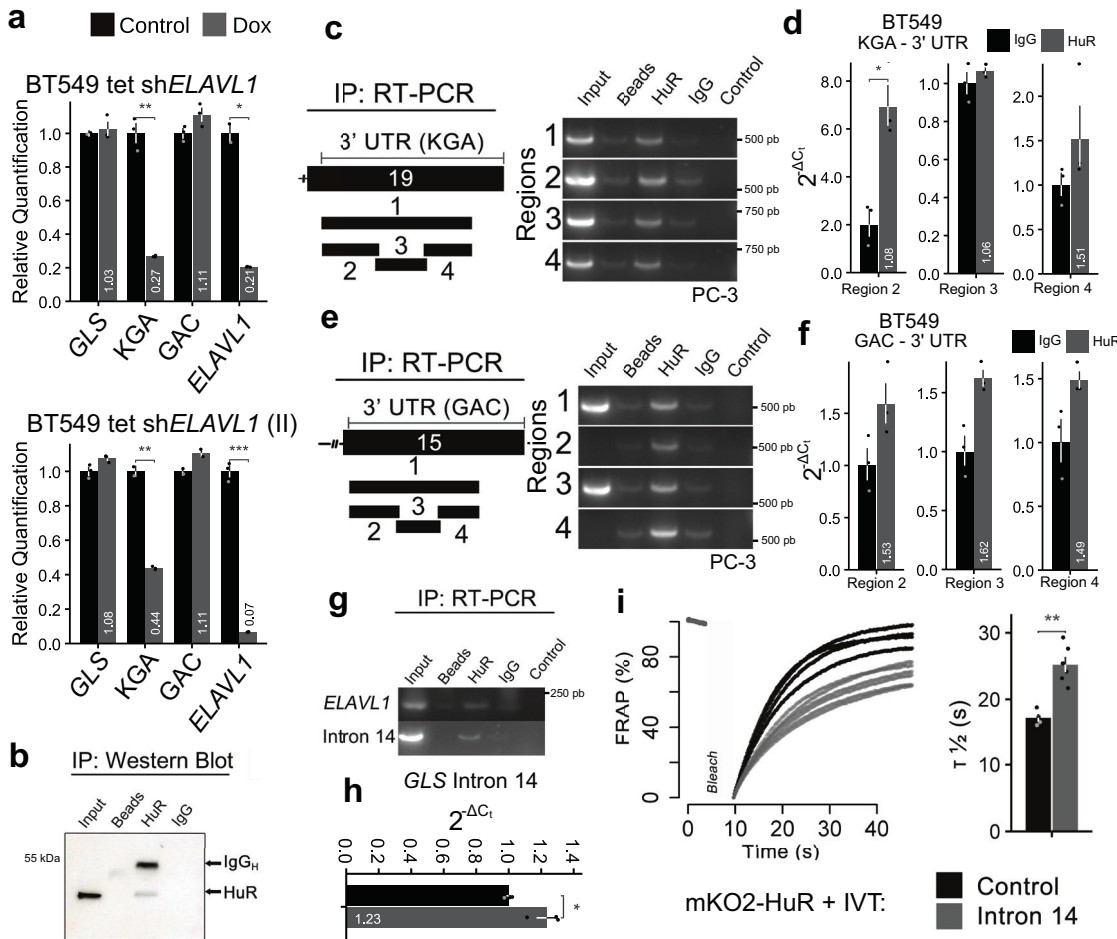

**Fig. 3 | HuR controls GAC and KGA mRNA levels in breast cancer cell lines and binds to *GLS* mRNA at multiple sites. a** Relative quantification of BT549 mRNA levels following doxycycline-inducible silencing of *ELAVL1*, using qPCR for two distinct shRNA sequences. **b** HuR was immunoprecipitated (western blot above, arrows indicate HuR and IgG heavy chain). The control IgG were used from rabbit, in opposing to the mouse anti-HuR antibody. Quantification of mRNA derived from 3′UTR KGA regions immunoprecipitated bound in HuR from (**c**) PC-3 cells using agarose gel and (**d**) from BT549 using qRT-PCR. Quantification of mRNA derived from 3′UTR GAC regions immunoprecipitated bound in HuR from (**e**) PC-3 cells using agarose gel and (**f**) from BT549 using qRT-PCR. Agarose gel from PC-3 (**g**) and q-RT-PCR from BT549 (**h**) revealed that HuR binds to its mRNA (as already published elsewhere[126]) in addition to *GLS* intron 14. **i** In vitro FRAP analysis of recombinant mKO2-HuR incubated with in vitro transcribed control RNA or intron 14. The bar plot (right) denotes T$_{1/2}$ recovery times. Statistical significance derived from Two-sided Welch's t-test (**a**, **d**, **f**, **h**, and **i**), error bars are SEM; qRT-PCR assays were evaluated in triplicate; otherwise, each point represents a replicate. *$p < 0.05$, **$p < 0.01$, ***$p < 0.0001$.

---

(from 1.74 to 1.22) and KGA (2.07 to 1.71) from polysome to monosome fraction, implying a reduction on translation of both isoforms. As expected, since HuR regulates its own locates to polysomes[55] and binds its own RNA[56], *ELAVL1* mRNA has also been moved to monosome fraction (Fig. 4e). Finally, using the AURA database[57], which describes multiple binding sites in the UTRs of the human genes that have been validated by photoactivatable ribonucleoside-enhanced crosslinking and immunoprecipitation (PAR-CLIP) experiments, we detected 3 and 24 HuR binding sites at the 3′-UTRs of the KGA and GAC transcripts, respectively (Supplementary Fig. 4c), confirming HuR direct binding to the 3′-UTR of GAC and KGA. Overall, we confirmed that HuR binds to intron 14 of *GLS*, favors the KGA isoform during splicing, binds to the 3′-UTRs of GAC and KGA, increases KGA and GAC's mRNA stability and potentially controls the translation of both isoforms.

Finally, to further characterize the interaction between HuR and intron 14, we evaluated the region around the acceptor splicing site of intron 14 using publicly available photoactivatable ribonucleoside-enhanced cross-linking and immunoprecipitation (PAR-CLIP) data[27,58]. Several HuR binding sites were found in the two evaluated cell lines (Fig. 5a). We then conducted evolutionary conservation evaluation (Fig. 5b) using VISTA database[59] and previous published data[9], as tool

to narrow down the specific HuR's binding region. The evaluation suggested a HuR's conserved binding region 159 bp upstream of intron 14's splicing acceptor site (Fig. 5c).

To confirm HuR's binding site on intron 14, we either mutated the AU-rich core or deleted an 86 bp long region containing this site on the RG6 splicing reporter (Fig. 5d). In BT549 cells, by either mutating ("Mutated") or deleting ("Deleted") the HuR conserved binding site, we significantly decreased the DsRed/eGFP (KGA/GAC) ratio by 10–14% compared to that in the control vector (compared between -Dox conditions; Fig. 5d), indicating a decrease in the preference for KGA formation when HuR's binding site on intron 14 was altered. *ELAVL1* knockdown further decreased DsRed/eGFP but to a smaller extent than that in the control (nonmutated/deleted vector), consistent with HuR being an important splicing regulator of intron 14 (Fig. 5d).

## HuR affects glutamine metabolism

Since HuR affects glutaminase isoform levels (favoring KGA over GAC), we evaluated its impact on glutamine metabolism. Dox-induced *ELAVL1* knockdown markedly increased BT549, MDA-MB-231 and Hs578t glutamine uptake (sh*ELAVL1*, Fig. 5e and Supplementary Fig. 8a), with a nonproportional increase in whole-cell lysate GLS

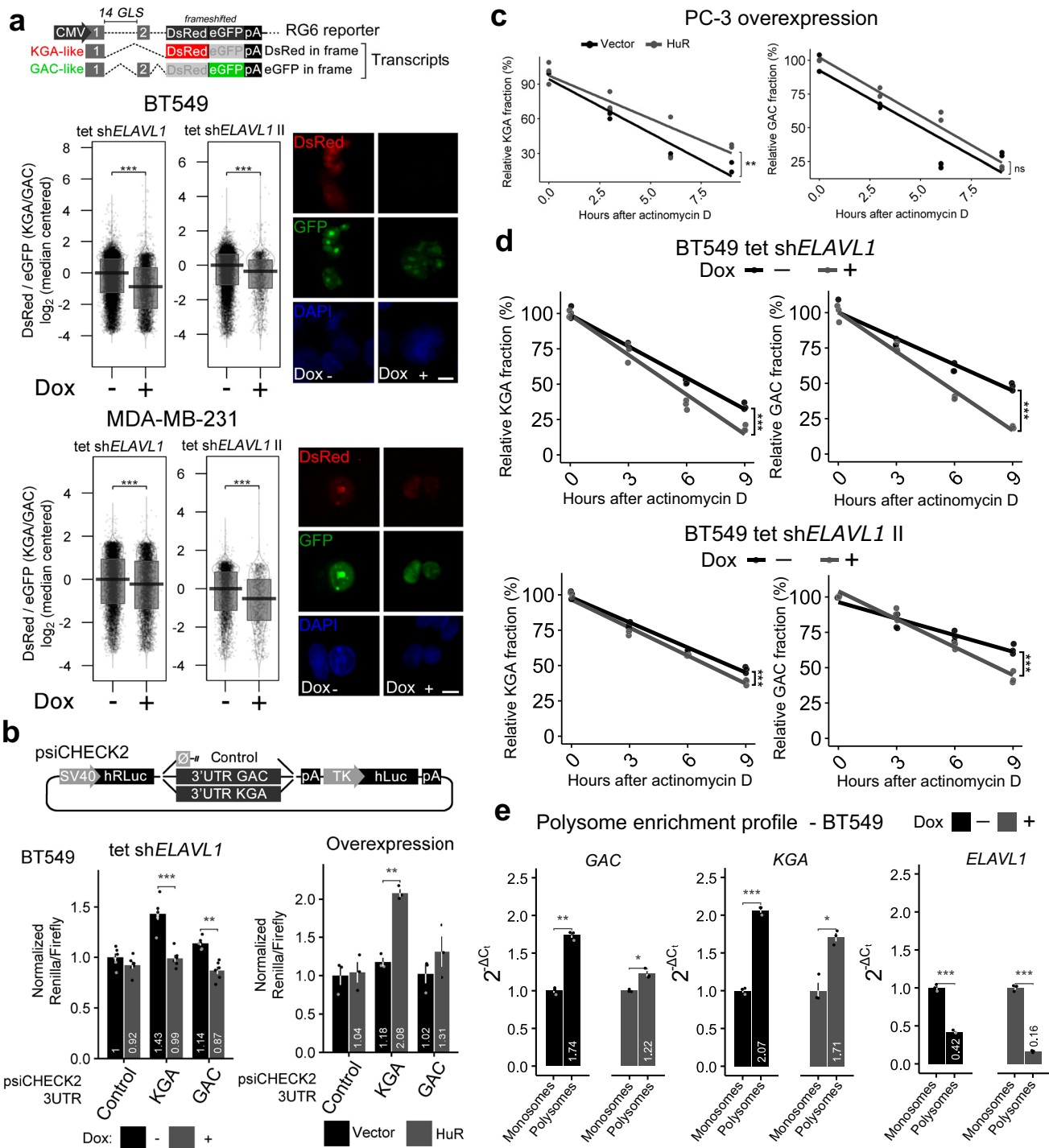

**Fig. 4 | HuR affects *GLS* alternative splicing and GAC and KGA mRNA metabolism. a**, top, Scheme of the RG6 splicing reporter for GLS intron 14[8]. HuR binding to intron 14 protects it from being spliced out and promotes an alternative splicing event that removes the reporter's exon 2 and generates DsRed in frame (producing DsRed protein, a surrogate for KGA's mRNA choice); HuR depletion promotes intron 14 splicing, leading to the retention of reporter's exon 2, which places the eGFP coding sequence in the correct frame (producing eGFP protein, a surrogate for GAC expression). Doxycycline-induced silencing of the *ELAVL1* cell line using sh*ELAVL1* I and sh*ELAVL1* II sequences in BT549 (**a**, middle; representative images are shown on the right) and MDA-MB-231 (**b**, bottom; representative images are shown on the right) cells led to a decrease in the DsRed/eGFP fluorescence ratio, indicating decreased KGA mRNA choice under this condition. At least 1158 independent cells evaluated for each group. **b**, top, Scheme of the psiCHECK2 reporter used to evaluate luciferase expression mediated by the GAC 3'-UTR or KGA 3'-UTR.

*ELAVL1* doxycycline-induced silencing decreased luciferase activity levels driven by both constructs but not the control (**b**, bottom left); HuR ectopic expression promoted the inverse effect (**b**, bottom right). At least 5 cell replicates for silencing and 3 for overexpression. **c** KGA (left) and GAC (right) mRNA decay after transcription blockage induced by actinomycin D incubation with PC-3 cancer cells (control cells or cells ectopically expressing HuR). **d** KGA (left) and GAC (right) mRNA decay after transcription blockage induced by actinomycin D incubation with BT549 cancer cells (control cells or *ELAVL1* silenced using doxycicline) using two distinct shRNAs. **e** RT-qPCR data for polisomal enrichment analysis following ultracentrifugation in sucrose gradient (*n* = 3). Each dot represents an individual cell or sample. Box plots represent the interquartile range; the vertical curve is the kernel density of the distribution, and the dark horizontal line denotes the median. Each dot represents an individual sample. Statistical significance was derived from Two-sided Welch's *t*-test, and error bars are SEM. *$p < 0.05$, **$p < 0.01$, ***$p < 0.0001$.

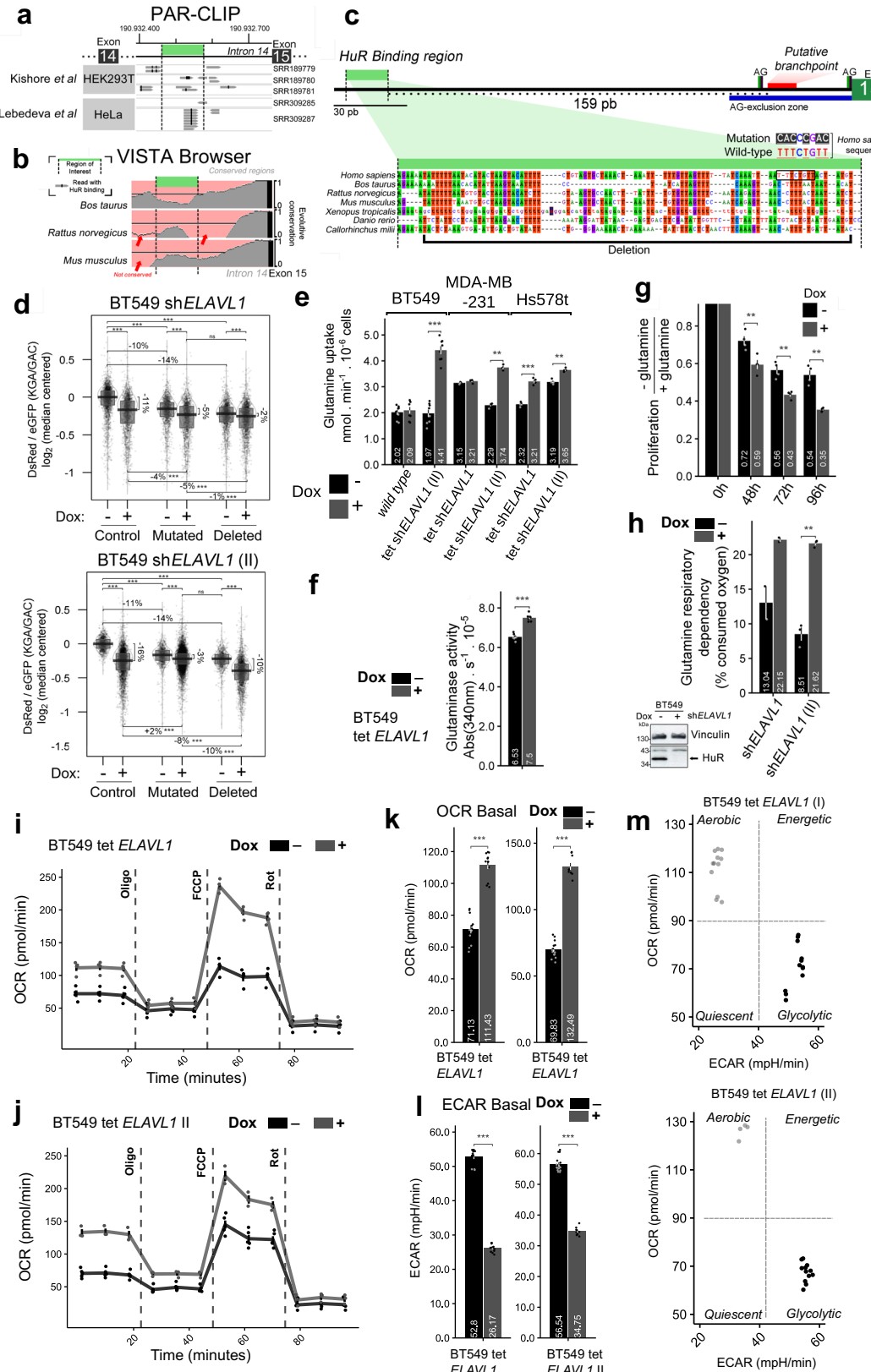

activity (inducible sh*ELAVL1*, Fig. 5f). This increase in glutamine uptake was also observed with inducible sh*ELAVL1* II in BT549 cells, inducible sh*ELAVL1* II in MDA-MB-231 cells, and inducible sh*ELAVL1* I and sh*E-LAVL1* II in Hs578t cells (Supplementary Figs. 5e and 8a). Doxycycline treatment by itself did not affect glutamine uptake (Fig. 5e, "wild type" condition). The same results were obtained for glutamine uptake

(Supplementary Fig. 8b) and GLS activity (sh*ELAVL1*, Supplementary Fig. 8c) after stable knockdown via sh*ELAVL1* and sh*ELAVL1* II in BT549 cells and via sh*ELAVL1* I in SKBR3 cells.

*ELAVL1* knockdown increased glutamine dependence for growth since glutamine withdrawal led to slower growth than that with complete media (Fig. 5g). Accordingly, the increase in glutamine uptake (and

**Fig. 5 | HuR binds to a conserved region within *GLS* intron 14 to promote KGA mRNA formation. a** PAR-CLIP-Seq experiments performed with HuR indicated a region proximal to the acceptor splicing site of intron 14 as the HuR binding region; the thick green bar indicates the region of PAR-CLIP in intron 14 with the maximum clustering of reads; (**b**) sequence conservation of the intronic region in *Mus musculus*, *Ratus norvegicus* and *Bos taurus*; a value of 1 on the y-axis represents the maximum similarity to the *Homo sapiens* sequence. (**c**) Scheme illustrating the distance between *GLS* intron 14 elements related to the splicing event (AG sites, AG-exclusion zone, and putative branchpoint) and the region identified by PAR-CLIP as a HuR binding site (HuR binding region, thick green bar); (**c**, bottom) multiple sequence alignment of the aforementioned sequence, indicating the mutation and deletion included in the RG6 splicing vector. Vertical dashed lines represent the identified region. **d** Mutation or deletion of the HuR binding site decreased the DsRed/eGFP fluorescence ratio compared to that in the control, indicating that HuR binding is important for the preference for KGA during splicing. Consistent with

HuR being a key element in the process, *ELAVL1* knockdown cells had a smaller variation in this ratio than control cells. The control is the wild-type sequence of intron 14. A total of 32,837 cells evaluated. Doxycycline-induced *ELAVL1* silencing in BT549, MDA-MB-231, and Hs578t cells increased glutamine uptake (**e**), glutaminase activity (**f**), and cell growth (**g**), and glutamine dependence on oxygen consumption (**h**), with representative blot for *ELAVL1* silencing. **i**, **j** OCR of BT549 cells silenced for *ELAVL1* using distinct shRNA target sequences on mitochondrial stress test (injections of oligomycin, FCCP, antimycin, and rotenone). Representation of basal OCR (**k**), ECAR (**l**), and dot-plot OCR vs ECAR (**m**) from **i** and **j**. Box plots represent the interquartile range; the vertical curve is the kernel density of the distribution, and the dark horizontal line denotes the mean. Each dot represents an individual cell, $n = 5$ for (**f**), $n > 2$ for (**h**), $n > 5$ for (**i–m**). Statistical significance was derived from ANOVA followed by Tukey's test (**d**, **g**) or Two-sided Welch's *t*-test (**e**, **f**, **h**, **k**, and **l**), and error bars are SEM. *$p < 0.05$, **$p < 0.01$, ***$p < 0.0001$. When indicated, 'sh*ELAVL1*' and 'sh*ELAVL1* II' refer to two distinct shRNA sequences.

glutamine activity) caused by *ELAVL1* knockdown was accompanied by a two-fold increase in glutamine dependency for oxygen consumption (Fig. 5h), implying that glutamine carbons were more relevant to the TCA cycle. HuR can shuttle between the nucleus and cytoplasm and has pleiotropic effects within the cell[60,61]. Moreover, this shuttling process can be governed by nutrient availability[62]. To check whether glutamine withdrawal consistently affected HuR cell location in BT549, MDA-MB-231, and Hs578t cells, we withdrew the nutrient and detected HuR localization by immunofluorescence. The results showed that glutamine withdrawal had different effects on HuR cellular location, either increasing (BT549) or decreasing (MDA-MB-231) or not affecting the fraction of HuR (Hs578t) in the cytoplasm (Supplementary Fig. 8d).

## HuR depletion leads to a shift toward aerobic metabolism

To further assess the effects of HuR on cellular respiration, we measured maximum and spare respiratory capacity. *ELAVL1* knock down increased maximum and spare respiratory capacity (Fig. 5i, j), pointing to an increased use of mitochondrial for respiration. *ELAVL1* knock down doubled basal oxygen consumption (Fig. 5k), while the proton efflux halved (Fig. 5l). This leads to the conclusion that by decreasing HuR levels, there is a shift from a more glycolytic to a more aerobic, glutamine-dependent, mitochondrial respiration (Fig. 5m). To evaluate the impact of *ELAVL1* knockdown alone or in combination with glutaminase inhibition with telaglenastat on glutamine metabolism, we performed GC-MS of metabolites extracted from BT459 cells kept for 6 h in media containing uniformly labeled $^{13}C^{15}N$-glutamine. A non-labeled, bulk metabolomic analysis (Fig. 6a, above) revealed that: 1. *ELAVL1* suppression affected the level of metabolites that concentrated in a cluster related to cAMP signaling (Fig. 6a, b, above); 2. Telaglenastat treatment affected the level of metabolites from central carbon metabolism and mTOR signaling (Fig. 6a, b, bellow); 3. *ELAVL1* suppression associated with Telaglenastat treatment enhanced the above-mentioned features and caused a shift between the levels of these two sets of metabolites, compared to control cells, indicating their capacity to act in synergy. Interestingly, *ELAVL1* knockdown did not affect total glutamine levels within the cells; however, as expected, telaglenastat treatment led to the accumulation of glutamine (Fig. 6c). Glutamate and α-ketoglutarate levels, on the other hand, were diminished after *ELAVL1* knockdown or telaglenastat treatment, and the effect was more pronounced with both *ELAVL1* knockdown and telaglenastat treatment combined (Fig. 6d, e).

While ~40% of the glutamate directly originated from the labeled glutamine (isotopologue m + 6, scheme in Fig. 6f) in both vehicle-treated (DMSO) and DMSO + Dox conditions, this percentage decreased to ~10% when cells were treated with telaglenastat in either -Dox or +Dox conditions (Fig. 6d, on the right). On the contrary, α-Ketoglutarate responded to the switch in the isoform levels caused by *ELAVL1* knockdown (as well as telaglenastat treatment) by non-significantly decreasing the relative amount of m + 5 isotopologue

from ~40% (DMSO) to ~20% (Fig. 6e, on the right). Telaglenastat combined with *ELAVL1* knockdown completely suppressed total α-ketoglutarate in the cells (Fig. 6e, on the left).

Glutamine metabolism in the TCA cycle can follow the oxidative and/or reductive carboxylation pathways depending on the cells' genetic alterations and environmental conditions[63–65] (scheme in Fig. 6f). In this sense, glutamine-derived isotopologues such as citrate m + 5, aspartate m + 3, malate m + 3, and fumarate m + 3 are indicative of α-ketoglutarate-isocitrate reductive reaction (with the incorporation of $CO_2$); on the other hand, glutamine-derived isotopologues such as citrate m + 4, fumarate m + 4 and m + 2 (second round of oxidation), malate m + 4 and m + 2 (second round of oxidation), as well as aspartate m + 4 (coming from oxaloacetate directly or indirectly produced from the TCA as a result of further citrate oxidation in the cytoplasm) and m + 5 (originating from transamination reaction using labeled amine from glutamate and labeled carbons from oxaloacetate) are indicative of oxidative glutaminolysis (Fig. 6f).

We verified that in the control condition (DMSO), the frequencies of both citrate m + 4 and citrate m + 5 were very similar (~15% of m + 4 and ~20% of m + 5; Fig. 6g, on the right), indicating that both directions of glutamine metabolism were taking place at similar rates. *ELAVL1* knockdown promoted a decrease in the percentage of both metabolites, although keeping them still at similar levels (~5–6%); telaglenastat treatment, on the other hand, sharply decreased overall citrate in the cells (Fig. 6g, on the left). There was more fumarate m + 4 (~40%) than m + 3 (~17%) in the DMSO condition, and in contrast to the effects on citrate, *ELAVL1* knockdown did not alter the percentages of these metabolites; however, telaglenastat treatment (and even moreso telaglenastat treatment combined with *ELAVL1* knockdown) decreased the levels of both metabolites (Fig. 6h). The equivalent was seen for aspartate (Fig. 6i). Malate m + 2, m + 3, and m + 4 isotopologues, unlike fumarate and aspartate, were profoundly affected by *ELAVL1* knockdown, which practically eliminated this metabolite from the cell (Fig. 6j); telaglenastat also caused a decrease in malate metabolites (Fig. 6j). Succinate isotopologues were poorly detected (Fig. 6k). In summary, glutamine metabolism through both *GLS* isoforms was essential for maintaining TCA cycle metabolite levels, since inhibition with telaglenastat decreased the overall pool of all evaluated metabolites. On the other hand, knocking down *ELAVL1* increased the GAC/KGA ratio levels and prompted various effects on the measured TCA cycle intermediates downstream of oxidative and reductive glutamine metabolism, implying that glutamine fate may change depending on which isoform processes the amino acid; moreover, combined *ELAVL1* knockdown and telaglenastat treatment uniformly intensified the effects of telaglenastat depletion of the measured metabolites. In conclusion, HuR controls the glycolytic-mitochondrial metabolism balance by favoring a more glycolytic cell state.

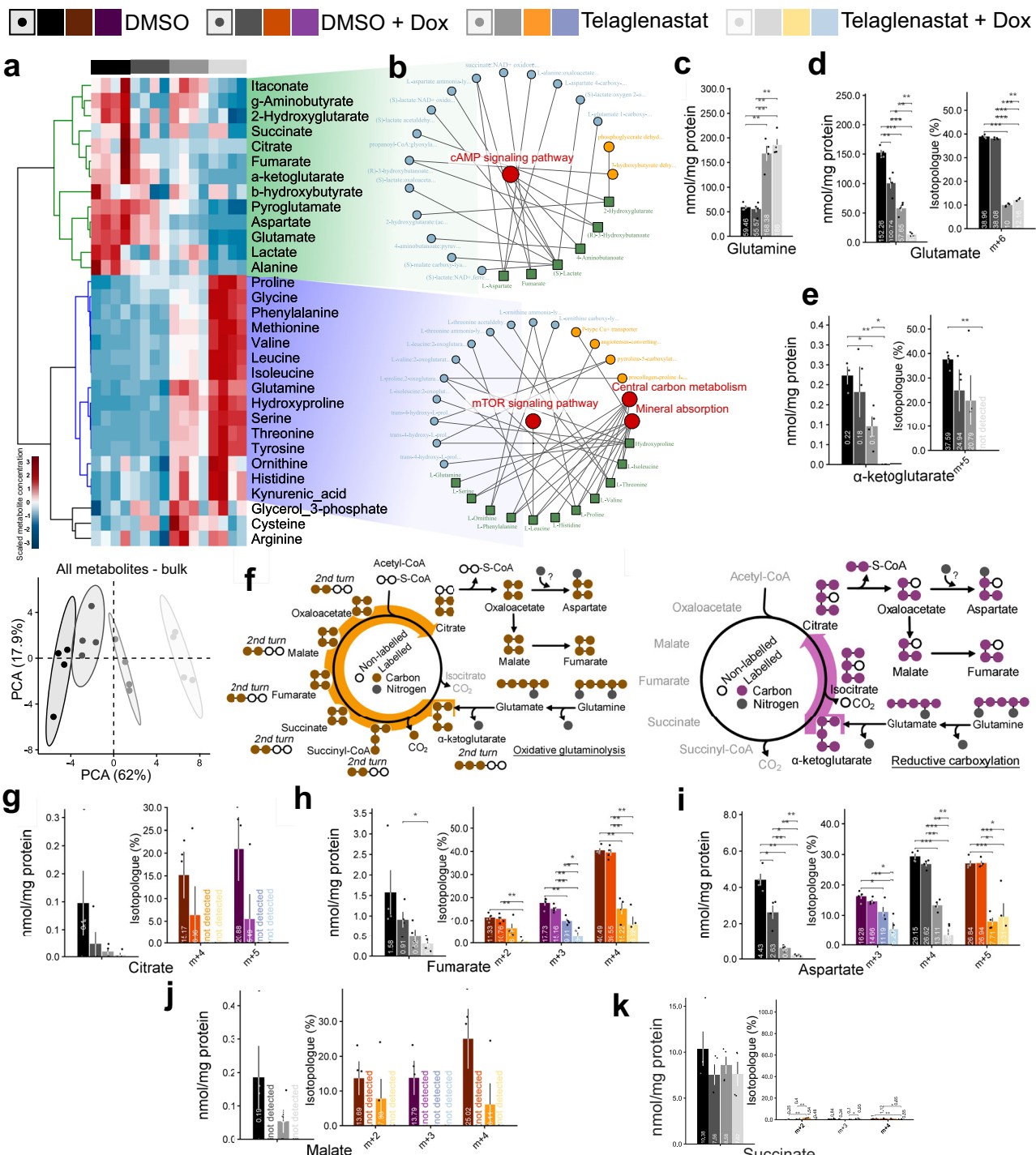

**Fig. 6 | HuR affects glutamine metabolism. a** Heatmap (above) from GC-MS metabolomics from Doxycycline-induced *ELAVL1* silencing in BT549 cells cluster evaluated compounds in (blue) telaglenastat responsive and (green) HuR and Telaglenastat responsive. (Below) Principal component analysis from same data, with 80% probability circles. **b** Network-based pathway enrichment using FELLA[113] for blue and green clusters. The effect of doxycycline-induced *ELAVL1* silencing on intracellular glutamine (**c**), glutamate (**d**), and α-ketoglutarate (**e**). **f** Scheme denotes glutamine reductive carboxylation (above) and oxidation (below) metabolic pathways and the related isotopologues. Thick arrows identify reductive pathway-related (purple) and oxidative pathway-related (brown) isotopologues. The effect of doxycycline-induced *ELAVL1* silencing on intracellular citrate (**g**), fumarate (**h**), aspartate (**i**), and malate (**j**), and succinate (**k**). For all mentioned letters, the overall pool is shown on the left and the isotopologue levels are shown on the right. Purple, brown and grays colors indicate isotopologues generated during reductive carboxylation and oxidative glutaminolysis and isotopologues common to both pathways, respectively. *n* = 4 for all experiments. Statistical significance was derived from Two-sided Welch's *t*-test; each dot represents an individual replicate, and error bars are SEM. *$p < 0.05$, **$p < 0.01$, ***$p < 0.0001$.

## *ELAVL1* knockdown sensitizes breast cancer cells to glutaminase inhibition

Since HuR was shown to affect glutaminase isoform levels, glutamine dependence for growth, and glutamine metabolism, we wanted to evaluate its impact on cell growth. *ELAVL1* inducible knockdown with sh*ELAVL1* I (Fig. 7a, above) and sh*ELAVL1* II (Fig. 7a, below) led to a significant decrease in BT549, MDA-MB-231 and Hs578t cell proliferation, which was not related to doxycycline treatment by itself

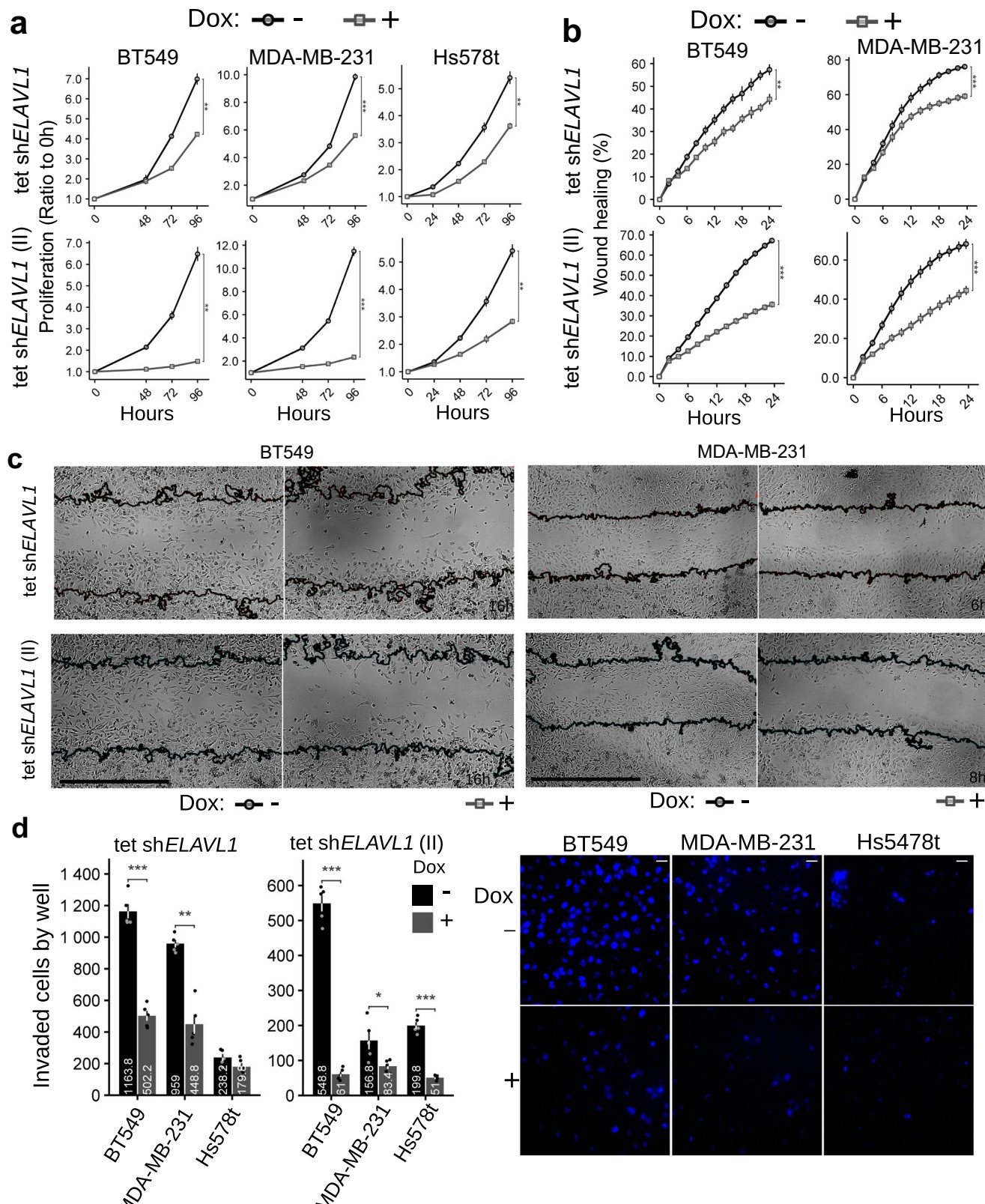

**Fig. 7 | *ELAVL1* knockdown decreases breast cancer cell growth, migration, and invasion.** Compared to the control, doxycycline-induced *ELAVL1* silencing decreased the proliferation of BT549 (**a**, left), MDA-MB-231 (**a**, middle), and Hs578t (**a**, right) cells; the migration, as measured by scratch healing rate, of BT549 (**b**, left), and MDA-MB-231 (**b**, right) cells was also decreased, representative images in **c**; cell invasion, as assayed by Matrigel-covered Boyden chamber assay, was also decreased in BT549, MDA-MB-231, and Hs578t (**d**, left), representative images in (**d**, right); in **c**, representative images below (scale bar represents 100 μm). In **d**, representative images of DAPI-stained nuclei located at the membrane. Statistical significance was derived from Two-sided Welch's *t*-test; each dot represents an individual replicate; otherwise, *n* = 4 for proliferation assays and *n* = 8 for migration/invasion assays; error bars are SEM. **p* < 0.05, ***p* < 0.01, ****p* < 0.0001. When indicated, 'sh*ELAVL1*' and 'sh*ELAVL1* II' refer to two distinct shRNA sequences.

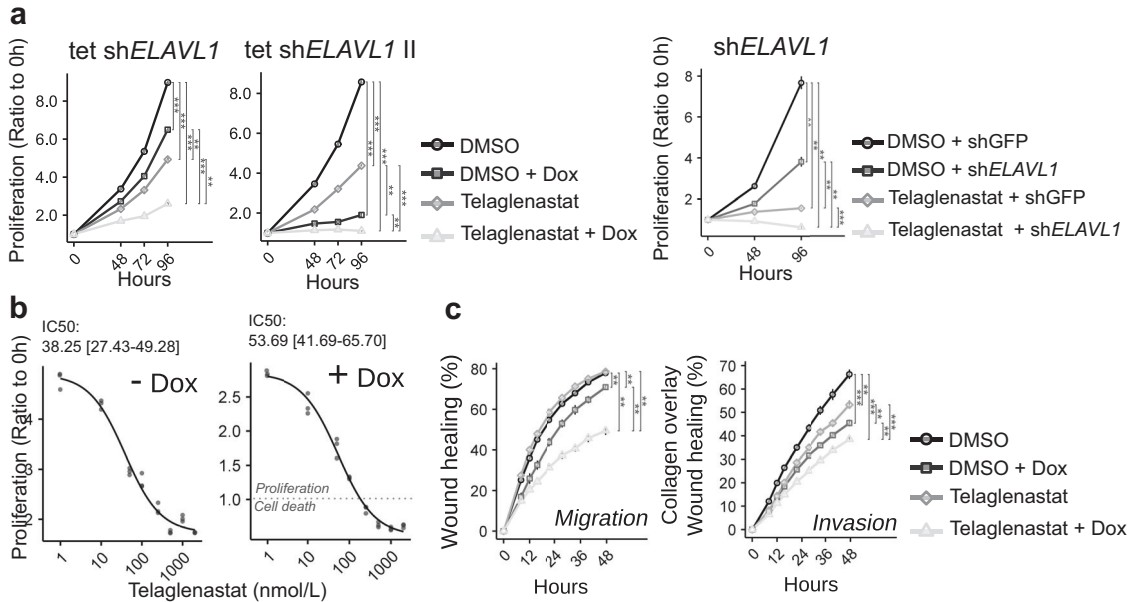

**Fig. 8 | ELAVL1 knockdown sensitizes breast cancer cells to glutaminase inhibition. a** Doxycycline-induced (left, sh*ELAVL1* and sh*ELAVL1* II) and stable (right) *ELAVL1* silencing in BT549 cells combined with GLS inhibition with telaglenastat (CB-839) decreased cell proliferatio, compared to that in control cells (DMSO or shGFP). **b** Growth response of BT549 to increasing amounts of telaglenastat (4-day treatment) with *ELAVL1* knockdown induced (+Dox, left) or not (−Dox, right). The IC50 value and 95% confidence interval (CI) are presented above the graphs. The gray dashed line indicates that there was a reduction in the final cell number compared to the number of seeded cells, denoting cell death. **c** Doxycycline-induced *ELAVL1* silencing of BT549 cells combined with GLS inhibition and telaglenastat decreased cell migration (left) and invasion (right) by 37 and 42%, respectively, compared to control cells (DMSO - Dox). Statistical significance was derived from Two-sided Welch's t-test; each dot represents an individual replicate; otherwise, $n = 3$ for proliferation assays and $n = 8$ for migration/invasion assays; error bars are SEM. $*p < 0.05$, $**p < 0.01$, $***p < 0.0001$.

(Supplementary Fig. 9a); in BT549 cells, stable *ELAVL1* knockdown also led to the same result (sh*ELAVL1* and sh*ELAVL1* II, Supplementary Fig. 9b). Additionally, stable *ELAVL1* knockdown slightly decreased BT549 tridimensional spheroid growth (Supplementary Fig. 9c).

Glutaminase is also essential to cancer cell aggressiveness, including cell migration and invasion processes[15,46,66]. Inducible *ELAVL1* knockdown in BT549, MDA-MB-231, and Hs578t cells promoted a significant decrease in cell migration (sh*ELAVL1* I, Fig. 7b, above; sh*ELAVL1* II, Fig. 7b, bellow, representative images in Fig. 7c; doxycycline did not affect migration by itself, Supplementary Fig. 10a) and invasion, as evaluated by collagen-overlaid wound healing assay (sh*ELAVL1* II for BT549 and sh*ELAVL1* I and sh*ELAVL1* II for MDA-MB-231, Supplementary Fig. 10c) and Matrigel-covered Boyden chamber assay (For BT549, MDA-MB-231, and Hs578t: sh*ELAVL1* is shown in Fig. 7d, left; sh*ELAVL1* II in Fig. 7d, middle; and representative images are displayed in Fig. 7d, right). Curiously, the effect of *ELAVL1* knockdown on the cell invasion events was more pronounced than that seen in the cell migration assays (1.8-4.1-fold decrease for the collagen-overlaid wound healing assay and 1.3-9.0-fold decrease for the Boyden chamber assay, compared to a 1.3–1.9-fold decrease in the wound healing assay, doxycycline-treated versus control).

Next, we evaluated the impact of glutaminase chemical inhibition with telaglenastat in conjunction with *ELAVL1* silencing on the proliferation of BT549 and MDA-MB-231 cells. Dox-induced and stable *ELAVL1* knockdown in BT549 (with sh*ELAVL1*) led to 28% and 50% decreases in growth at 96 h compared to that in the control (DMSO or DMSO+shGFP); conversely, telaglenastat treatment alone decreased cell growth by 45% and 80% at 96 h compared to that in the control (DMSO or DMSO+shGFP, respectively) (Fig. 8a, above and below, respectively). Combined Dox-induced and stable *ELAVL1* knockdown and telaglenastat treatment promoted a further decrease in cell growth, with 71% and 92% growth inhibition at 96 h compared to that in the control (DMSO or DMSO+shGFP) (Fig. 8a, above and below, respectively). In line with these findings, a telaglenastat (and the

mechanistically related glutaminase inhibitor BPTES[6]) cell proliferation dose-response assay showed that Dox-induced *ELAVL1* knockdown sensitized cells to glutaminase inhibition, as evidenced by pronounced cell death at higher telaglenastat or BPTES doses in doxycycline-treated cells (but not control cells, -Dox) (Fig. 8b and Supplementary Fig. 11a, respectively). Moreover, combined Dox-induced *ELAVL1* knockdown and telaglenastat treatment promoted a further decrease in cell migration (Fig. 8c, left) and invasion (Fig. 8c, right), with 37% and 42% migration and invasion inhibition, respectively, at 48 h of treatment compared to that in the control (DMSO). On the other hand, in MDA-MB-231 cell line, telaglenastat monotherapy by itself showed a profound effect on proliferation, migration, and invasion of the cells, with dox-induced *ELAVL1* silencing marginally contributing with a further effect when performed in combination, especially for the migration of the cells (Supplementary Fig. 11a, b, respectively). Equally, telaglenastat and BPTES IC50 values did not change with *ELAVL1* silencing (Supplementary Fig. 11c).

We then evaluated the relative contribution of each glutaminase isoform to BT549 cell growth. Selective knockdown of GAC did not affect cell growth and invasion compared to that in the control (shLuc) (Supplementary Fig. 12b, western blot on the left and graphs on the right); knocking down *ELAVL1* alone noticeably decreased proliferation, migration, and invasion, but combining *ELAVL1* and GAC knockdown (which depleted both glutaminase isoforms) did provide a small further decrease in cell growth, migration and invasion compared to that with *ELAVL1* knockdown alone.

Finally, we restored KGA after knocking down *ELAVL1* by expressing ectopic KGA in the cells. Although KGA ectopic expression (KGA-V5) increased glutaminase activity (Supplementary Fig. 12c, western blot on the left and graph on the right), it did not provide any further cell growth (and migration) capability compared to that in control (Mock -Dox) and doxycycline-treated cells (Mock +Dox) (Supplementary Fig. 12d, left and right, respectively). Taken together, we conclude that HuR is important for breast cancer cell growth, migration, and

invasion. HuR depletion sensitizes cells to glutaminase inhibition; however, the action of HuR on cell growth/migration cannot be explained only by glutaminase isoform switching, implying that pleiotropic roles, other than glutamine metabolism effects, are also involved.

## Discussion

RNA binding proteins regulate all steps of RNA life, influencing its processing, modification, stability, translation, and localization, with profound effects on cell behavior[67,68]. HuR binds to AU-rich elements scattered throughout the genome, especially at the RNA 3′-UTR and intronic regions, regulating many genes and cellular processes[69]. HuR is also involved in several aspects of cancer cell transformation and progression and is considered a promising drug target for treating cancer[22,36], since it correlates with high-grade malignancy and poor prognosis in various tumors[22]. However, the role of HuR in regulating glutamine metabolism in cancer remains elusive.

Despite HuR's well-characterized role in controlling RNA stability and translation[36,70,71], it was only recently recognized as an alternative splicing regulator[27,28,72–74], and few targets have been experimentally evaluated[75–78]. In this manuscript, we investigated the relevance of HuR in controlling glutamine metabolism in breast cancer by regulating *GLS* mRNA metabolism. We started with a TCGA pancancer evaluation of *ELAVL1* overexpression in multiple tumors compared to their paired normal tissues. *ELAVL1* was overexpressed in many tumors, including kidney chromophore, rectum and colon adenocarcinomas, lung and cervical squamous cell carcinomas, and breast cancer carcinoma, confirming previous data from the literature[29,79]. In breast cancer, *ELAVL1* high levels were correlated with a worse prognosis; these results corroborated a previous study[39] but contradicted other former works[80,81]. Coherently, *ELAVL1* is more expressed in more aggressive BRCA subtypes, such as the triple negative one, the PAM50 basal subtype, and the invasive ductal carcinomas (Supplementary Fig. 13a–f). Although *ELAVL1* expression level is at the highest in the basal subtype, this is the subtype where we measured the lowest KGA/GAC mRNA ratio. On the other hand, the highest KGA/GAC mRNA ratio was measured on the Luminal A/B and HER2+ subtypes. Our data revealed that HuR has the most decisive impact on the KGA/GAC protein level on a cell background that is glutamine withdraw and glutaminase-inhibition resistant (which even includes some TN tumor cells), implying that it is not the HuR levels by itself, but glutamine-dependent TCA cycle itself is the critical factor behind HuR having the most pronounced effect over KGA/GAC levels. The reason for that, we believe, is because cells highly dependent on glutamine for TCA cycle anaplerosis utilize HuR-independent mechanisms to enhance GAC levels.

By using publicly available RIP-Seq, RNA-seq and PAR-CLIP-Seq transcriptomic data, we showed, for the first time, that HuR binds to *GLS* mRNA at intron 14 (the intron before exon 15, which is the site of splicing that generates the KGA isoform instead of the GAC isoform) and at the 3′-UTRs of GAC's and KGA; in addition, *ELAVL1* knockdown in HeLa cells displayed an alteration in exon 15 (GAC-exclusive) and exon 16-19 (KGA-exclusive) levels. In breast cancer cell lines, knocking down *ELAVL1* changed the balance between KGA and GAC protein and mRNA levels, with a decrease in the first followed by a not-always proportional increase in the second. HuR immunoprecipitation followed by RT-PCR confirmed that HuR binds to intron 14 of *GLS* and GAC and KGA 3′-UTRs; a luciferase reporter assay showed that luciferase genes regulated by both GAC and KGA 3′-UTRs have enhanced expression under HuR ectopic expression. On the other hand, a stability assay followed by actinomycin treatment showed that HuR enhanced KGA (but not GAC) mRNA stability, in opposition to the reduction of both isoform's stability under *ELAVL1* suppression. HuR potentially increases the translation of GAC and KGA, as indicated by the polysome/monosome assay. An alternative splicing assay with a

fluorescent reporter also revealed HuR's role in favoring the formation of KGA mRNA over GAC mRNA. Altogether, these data reveal the particular importance of HuR in controlling several aspects of GLS mRNA metabolism and protein isoform levels in breast cancer. Accordingly, in breast tumors, enhanced *ELAVL1* levels correlate with an increase in both GAC mRNA levels and KGA mRNA levels. HuR-mediated effects on *GLS* RNA metabolism are summarized in Supplementary Fig. S14.

Knocking down *ELAVL1* in the cell lines revealed two different scenarios: in the cell lines where GAC and KGA were expressed in a more balanced way (two bands with similar intensities clearly seen by using an antibody that recognize both proteins; SKBR3 and MDA-MB-157), *ELAVL1* knockdown caused a shift in the isoform proportion, with a clear increase in GAC and a subsequent decrease in KGA, which reached undetectable levels; in the cell lines where GAC was already the more prominent band (BT549, MDA-MB-231, HCC38 and Hs578t), knocking down *ELAVL1* caused a visible decrease in the already faint KGA band followed by a not always apparent increase in GAC. We speculate that in this second group of cells, which are recognizably more dependent on glutamine and glutaminase for growth and survival[46], other mechanisms co-occur to stabilize GAC levels since this isoform is the more active glutaminase[12,82]. An extended analysis in a panel of 12 breast cancer cell lines showed that HuR, KGA, and GAC proteins are ubiquitously present (Supplementary Fig. 15a), with a slight positive correlation between HuR and KGA band intensity levels (Supplementary Fig. 15b). We also found a positive correlation between the mRNA levels of *ELAVL1* and KGA and *ELAVL1* and GAC (Supplementary Fig. 15c).

*ELAVL1* silencing in breast cancer cells affected glutamine metabolism, leading to an increase in glutamine uptake, glutaminase activity, and glutamine dependence for growth. Moreover, under HuR suppression, cells displayed enhanced oxygen consumption related to the oxidation of glutamine carbons in the TCA cycle and are shifted to a more aerobic profile. A targeted metabolomic assay performed with uniformly labeled glutamine confirmed HuR's direct link to glutamine catabolism in the TCA cycle: decreased HuR levels decreased glutamine-derived glutamate, as well as citrate and malate, in both reductive and oxidative glutamine metabolic pathways. Curiously, the isotopologue fractions of fumarate and aspartate, the latter of which is provided by the glutamate-oxaloacetate transaminase (GOT) 1 and 2 enzymes, did not change upon *ELAVL1* knockdown, despite the total metabolite levels reduction seen (Fig. 6a). These results, combined with the direct correlation of the expression levels of *ELAVL1* with the genes of the malate-aspartate shuttle, as seen with the breast cancer transcriptome data, suggest that while citrate and malate are likely more dependent on KGA metabolism, fumarate and aspartate can be generated by glutamine metabolism from either isoform. Whether or not (and how) glutamine-glutamate metabolism of the different isoforms dictates the fate of the metabolites further down the TCA cycle needs further investigation, as well as the potential preferential role of GAC in glutamine dependency over KGA.

HuR depletion caused an increase in the more catalytically active enzyme (as determined in vitro[12,82]), GAC. Although cells capture more glutamine from the media under *ELAVL1* knockdown, intracellular glutamine did not change, and there was a decrease in glutamate levels compared to those in the control condition. Such a situation may indicate that, under HuR depletion conditions, in addition to having increased glutaminase activity (dictated by higher levels of a more active glutaminase), cells also present increased shuttling of glutamine carbons into the TCA cycle, which may be a result of the alteration in the activity of other enzymes. Such an alteration may be the consequence of a pleiotropic effect of HuR on cell metabolism and may explain the increased dependence of cells on glutamine for growth, but the precise mechanism behind that needs further clarification. Transcriptomic evaluation of TCGA breast tumor samples with high

*versus* low *ELAVL1* levels revealed that several genes from the glucose, TCA, fatty acid synthesis, beta-oxidation, and urea cycle pathways had altered expression levels (Fig. 1e, Supplementary Data 4). Whether HuR directly affects the transcription levels of these genes and the exact mechanism underlying this effect requires further study. Regardless, all the measured TCA metabolites were deeply affected by glutaminase inhibition with telaglenastat, an effect that was potentiated by the combination with *ELAVL1* knockdown. Such a combination reveals a therapeutic opportunity for breast cancer.

We[46] and other groups[83,84] have already proposed that glutaminase-targeted therapies can be combined with other targeted therapies to seek improved benefits. Some HuR inhibitors are already known; for example, the MS-444 compound inhibits HuR dimerization and export from the nucleus, inhibiting its cytoplasmic roles[22]. Newer molecules derived from tanshinone block the binding of HuR's first and second RNA (but not the third) recognition binding motifs to RNA[85,86]. Since the known inhibitor only partially inhibits the HuR complex role in the cell, other works have also proposed genetic-based therapies, as well as in vivo delivery of silencing elements[87,88]. The glutaminase inhibitor telaglenastat is under phase I/II of clinical trials for several solid and hematopoietic tumors, including breast cancer[89]. The benefit of combined glutaminase and HuR inhibition with small molecules is an exciting proposition but lacks further in vitro and in vivo proof.

In conclusion, we showed that HuR is a crucial regulator of glutaminase RNA metabolism in breast cancer that affects different aspects of mRNA metabolism, such as mRNA stability and splicing. Overall, HuR coordinates glutamine metabolism through the TCA cycle, and its depletion renders cells more dependent on glutamine for growth and migration and more sensitive to glutaminase inhibition. These findings present a therapeutic opportunity for breast cancer.

## Methods

### Antibodies, plasmids, and reagents

Primary antibodies used for immunofluorescence and immunoblotting were Vinculin (Abcam ab18058), Actin (Abcam ab3280), GLS (Abcam ab156876), GAC (RheaBiotech IM-0322), KGA (RheaBiotech IM-0519), HuR (Molecular Probes mp21277 and Cell Signaling 12582), and V5 (Life Technologies 46-1157). pLKO.1 puro was a gift from Bob Weinberg (Addgene plasmid # 8453); pLKO.1-blast, used in GAC silencing experiment was a gift from Keith Mostov (Addgene plasmid # 26655); Tet-pLKO-puro was a gift from Dmitri Wiederschain (Addgene plasmid # 21915); psPAX2 was a gift from Didier Trono (Addgene plasmid # 12260); pMD2.G was a gift from Didier Trono (Addgene plasmid # 12259); pUMVC was a gift from Robert A. Weinberg (Addgene plasmid # 8449); pCMV-VSV-G was a gift from Robert Weinbeg (Addgene plasmid # 8454); pQC mKorange IX was a gift from Connie Cepko (Addgene plasmid # 37344), previously modified by replacing the mKO2 coding sequence with an empty MCS[7], generating pQC MCS IRES Puro (Addgene # 110343) and pQC V5 MCS IRES Puro (Addgene # 110342). For KGA rescue experiment, the used plasmid was previously altered to confer geneticin resistance[7], using neo (neomycin phosphotransferase) coding sequence extracted from pQCXI Neo DsRed-LC3-GFP, a gift from David Sabatini (Addgene plasmid #31183), generating pQC MCS IRES G418 (Addgene # 110344) and pQC V5 MCS IRES G418 (Addgene # 110345); pX330 was a gift from Feng Zhang (Addgene plasmid # 42230); RG6 Intron 14 GLS reporter vector was previously published[8]. For BT549 CRISPR Knock-in experiment, lentiCas9-Blast (Addgene plasmid #52962) and lentiGuide-Puro (Addgene plasmid # 52963), both gifts from Fenz Zhang, were used. All the reagents were purchased from Sigma-Aldrich unless otherwise stated.

### Bioinformatic analysis

TCGA data were obtained from the Genomic Data Commons portal. The gene-level expression was defined in upper-quantile FPKM units,

isoform-level expression was defined in RPKM units of UCSC isoforms in *legacy* TCGA pipeline. For body-map representation, data was plotted using QGIS[90] and R-packages maptools[91], ggplot2[92], and gpclib[93]. For progression-free interval[38] *p*-value minimization[37,94], R-packages survival[95] and survminer[96] were used. Briefly, all possible cutoffs dividing patients into two groups were evaluated by log-rank tests, and the division with the smallest *p*-value was used. RIP-Seq data was obtained from ENCODE project[47] (SRR504455-6 and SRR504447-8 for GM12878; SRR504459-60 and SRR504453 for K562), RNA-Seq and PAR-CLIP-Seq data for *ELAVL1* silencing in HeLa cells were obtained from Lebedeva et al.[27]. PAR-Clip-Seq data from HEK293T were obtained from Kishore et al.[58]. RNA-Seq data for *ELAVL1* knockout in MIA PaCa-2 cells were obtained from McCarthy et al.[48]. Quality was evaluated with FastQC[97]; sequence trimming was performed by Skewer[98] following published guidelines[99], sequences were aligned to GRCh38 using STAR[100,101] (2-pass mode). For RIP-Seq analysis, each GENCODE[102] (v25) transcript without CCDS[103] entry was removed, retaining the only gene with two or more transcripts (exception for genes with transcripts varying only for transcriptional start sites, which were removed from the analysis). This new annotation was used to detect introns around exons varying among the isoforms, and HTSeq[104] was used to count the aligned reads over those introns. Since no replicates were available, reads were subsampled in a bootstrap-like approach, as previous proposed[105], and counts used for DESeq2[106] analysis. For RNA-Seq analysis of HeLa si*ELAVL1*, original GENCODE annotation was used, and DEXSeq[107] evaluated differential exon usage, with the same bootstrap-like approach proposed[105]. Gene ontology enrichment was evaluated using Go.db[108] and goseq[109]. Aligned PAR-CLIP-Seq data was analyzed with IGV[110]. The evolutionary conservation of intron 14 was described with VISTA Browser[59] and previously obtained sequences[9]. RNA-Seq analysis of TNBC cell lines was performed as described previously[45]. Projections from TCGA were performed using singscore[111] and entire MSigDB[112], metabolic signatures were defined based in a regex search for "metaboli" ignoring case. Network-based metabolome enrichment from selected clusters were performed using FELLA[113], factoextra and igraph r-packages. All plots, otherwise stated, were created using ggplot2, yarrr or base functions from R.

### Cell culture

BT549 (ATCC HTB-122), Hs578t (ATCC HTB-126), SKBR-3 (ATCC HCB-30), MDA-MB-157 (ATCC HTB-24), MDA-MB-231 (ATCC HTB-26), MDA-MB-436 (ATCC HTB-130), MDA-MB-453 (ATCC HTB-131), MDA-MB-468 (ATCC HTB-132), HCC38 (ATCC CRL-2314), HCC70 (ATCC CRL-2315), HCC1806 (ATCC CRL-2335), HCC1937 (ATCC CRL-2336), and PC-3 cell lines were cultivated in RPMI 1640 media (Sigma) supplemented with 10% fetal bovine serum (Vitrocell). HEK293T (ATCC CRL-3216) and BALB/3T3-A31 (ATCC CCL-163) cell lines were cultivated in DMEM high glucose media (Sigma) supplemented with 10% fetal bovine serum (Vitrocell). All cell lines were obtained from the American Type Culture Collection (ATCC) in March of 2014, with the exception of SKBR3, which was purchased in May of 2010. HEK293T was a gift from Dr. Ângela Saito, and BALB/3T3-A31 was a gift from Dr. Daniel Maragno Trindade. All cells were grown at 37 °C in a 5% $CO_2$ humidified incubator. Mycoplasma testing was routinely performed by PCR using the primers sequences GPO-3 (5′-GGGAGCAAACAGGATTAGATACCCT-3′) and MGSO (5′-TGCACCATCTGTCACTCTGTTACCCTC-3′) which amplifies the rRNA of 16 different mycoplasma species. Positive cell cultures were discarded or treated with Plasmocin (InvivoGen). All the cellular assays were performed one to three times.

### Viral production and cellular infection

Viral particles were produced using HEK293T cells transfected using Polyethylenimine (Polysciences 23966-2). The ratio of packing vectors was 4:3:1 of transfer:psPAX2:pMD2.G for lentiviral particles or transfer:pUMVC:pCMV-VSV-G for gammaretrovirus. For low-titer tet-pLKO

vectors, particles were PEG-concentrated. Before use, all productions were titred using BALB/3T3-A31 cells, and the transduction performed using 0.3 multiplicity of infection (MOI). Cells positive for insertion were selected using the appropriate agent, puromycin (1 µg/mL, for both selection and maintenance, Thermo-Fisher), geneticin (1000 µg/mL for selection and 200 µg/mL for maintenance, Sigma), and blasticidin (2 µg/mL, for both selection and maintenance, Thermo-Fisher). Sequences for silencing were TRCN0000276129 (sh*ELAVL1* I), TRCN0000276186 (sh*ELAVL1* II), 5′CCTCTGTTCTGTCAGAGTT3′ (shGAC[82]), TRCN0000072197 (shGFP), and 5′CTTACGCTGAG-TACTTCGA3′ (shLuc[114]). For silencing induction, 50 µg/mL doxycycline hyclate was added 48 h before experiment preparation.

### Proliferation assays

Two thousand cells were plated in 96-well plates and allowed to adhere overnight. For the glutamine deprivation assay, complete cell-culture was removed, and cells were washed with PBS and then incubated in RPMI lacking glutamine (Vitrocell) supplemented with 10% charcoal-stripped FBS (Invitrogen), for all experiments media was replaced every 48 h. For spheroid proliferation assay, four thousand cells were plated in ultra-low attachment plates (Corning #4515), and half of the culture media was replaced every 48 h. Cells were fixed with 4% methanol-free paraformaldehyde in phosphate buffer and stained with 10 µg/mL DAPI in PBS with 0.2% Triton X-100. In the glutaminase inhibition proliferation assay, media with 1 µM CB839 (Selleckchem, S7655) or 0.01% DMSO were added to the cells. Plates were imaged in fluorescence microscope Operetta (Perkin Elmer) and the total number of DAPI-stained nuclei counted using Columbus software (Perkin-Elmer). The number of nuclei at a specific time point was normalized against the number of seeded cells (seeding control was collected ~12 h after seeding and thus considered timepoint 0 h).

### Quantitative PCR

Procedures for gene expression quantification were performed as previously described[7]. RNA was extracted from samples using the TRI Reagent following the manufacturer's instructions (Sigma). Complementary DNA synthesis was performed with GoScript™ Reverse Transcriptase (Promega) using random hexamers and (dT)18 mixture (7:5) following the manufacturer's instructions. PCR amplification was performed with Power SYBR Green PCR MasterMix (Applied Biosystems) as instructed by the manufacturer. Samples were run on the Applied Biosystems 7500 real-time PCR system and analyzed following the 2-ΔΔCT method. The following primers were used: rRNA18S (5′ATTCCGATAACGAACGAGAC3′ and 5′TCACAGACCTGTTATTGCTC3′), *ELAVL1* (5′CATTAAGGTGTCGTATGCTC3′ 5′CTGGACAAACCTGTAGTCTG3′), *GLS* (5′AAAGCAGTCTGGAGGAAAGG3′ and 5′AGTAGAATGCCTCTGTCCATCTA3′), GAC (5′GATCAAAGGCATTCCTTTGG3′ and 5′TACTACAGTTGTAGAGATGTCC3′), KGA (5′TGGTGATCAAAGGGTAAAGTC3′, and 5′TGCTGTTCTAGAATCATAGTCC3′). Total RNA was extracted from patient tissues from a Romania cohort[7] using TRI reagent following the manufacturer's protocol. Reverse transcription was performed using random hexamers with SuperScript III Reverse Polymerase according to the manufacturer's protocol (Invitrogen). Quantitative RT-PCR analysis was performed with SYBR Green SuperMix following manufacturer's protocol (Thermo-fisher). The following primers were used: *ELAVL1* (described above), KGA (described above), *GAC* (described above), and *TBP* (5′GCACAGGAGCCAAGAGTGAA3′ and 5′TCACAGGTCCCCACCATATT3′). *TBP* primers were used as internal controls. Relative expression levels were calculated as -ΔCt.

### RNA immunoprecipitation, mRNA stability, and polysome fractioning

HuR immunoprecipitation and characterization of cargo RNA was performed as described elsewhere[76,115,116]. Briefly, cells were lysed (20 mM Tris-HCl pH 7.4, 150 mM NaCl, 1% Triton X-100, 0.5 U/µL

RNaseOUT, 1 X Roche Protease Inhibitor Cocktail), incubated with Anti-HuR (Molecular Probes, mp21277), sonicated to disrupt RNA, and bound to BSA/Salmon Sperm blocked Magnabeads (Invitrogen) for 16 h at 4 °C. Beads were washed four times with NT2 buffer (50 mM Tris-HCl pH 7.4, 150 mM NaCl, 1 mM MgCl₂, 0.05% NP-40) and RNA extracted and quantified as previously described[76,115,116]. For HEK293T assays, a control IgG were used from rabbit, in opposing to the mouse anti-HuR antibody. To BT549 assay, a mouse IgG were used as control. Primers used were Intron 14 pre-mRNA (5′AAGAATTATGCAAAACCAACAG3′ and 5′TATGGATCCTAGTTGTTCAAGCAACATACATAAT3′), KGA 3′ UTR region 1 (5′GGTCTCAAATCCCAAGATTTAAAT3′ and 5′TGAAGCTAGGGTGAGAGAGAGACA3′), region 2 (5′TGTCTCTCTCTCACCCTAGCTTCA3′ and 5′AATCTGGAATGATCCAGTGGTCCC3′), and region 3 (5′GGGACCACTGGATCATTCCAGATT3′ and 5′CACAAAGCGGCTGCTCTTTGAAT3′); GAC 3′ UTR region 1 (5′GAAATGGGTTCTAGTTTCAGAATG3′ and 5′ACTCTGACAGAACAGAGGAGTTGC3′), region 2 (5′GCAACTCCTCTGTTCTGTCAGAGT3′ and 5′GGAAGAAGGAAGAAGTGTGAATAGGTCC3′), and region 3 (5′GGACCTATTCACACTTCTTCCTTCTTCC3′ and 5′CCAATTAAGGCATTCGGTTGCCCA3′). The mRNA stability assays were performed supplementing medium with Actinomycin D (Sigma-Aldrich) at 5 µg/mL (PC-3) or 1 µg/mL (BT549) and performing RNA extraction using TRI Reagent (Sigma-Aldrich) at indicated timepoints. Polysome fractioning were performed using hand-made 10%-50% sucrose gradient[117] with the conditions described by Poria & Ray[118] and a protease inhibitor cocktail previously described[45]. Centrifugation performed using an Optima LE-80K (Beckman-Coulter) with an SW 41 Ti rotor (r_{max} of 160.000 g at 30.000 rpm) at 4 °C for 4 h. Fractioning was manually performed piercing the tubes in the bottom part and gravity-flowing them into 96-well microplates. Absorbance reading realized using a Nano-Drop2000c (Thermo-Scientific). Fractions were combined to generate monosome and polysome regions.

### Western blot and immunofluorescence

Procedures for western blot were previously described[45]. Briefly, protein lysates were resolved in 4-20% gradient SDS-PAGE and semy-dry electroblotted against PVDF membranes, using six WypAll X60 (Kimberly-Clark) soaked in alcohol-free buffer[119] in each sandwich side, at 0.325 mA/mm². Membranes were blocked in 3% non-fat dry milk diluted in tris buffered saline with 0.05% Tween 20, incubated with primary antibodies overnight at 4 °C and probed with HRP-conjugated secondary antibodies (Sigma), before detection using SuperSignal Wes Pico Chemiluminescent Substrate (Pierce) (protocols.io https://doi.org/10.17504/protocols.io.puxdnxn). Procedures for immunofluorescence microscopy were previously described[46].

### Fluorescence recovery after photobleaching

Run-off in vitro transcription (IVT) of *GLS* intron 14 or empty vector were performed using Riboprobe in vitro transcription system (Promega), from sequences cloned into pGEM-T easy/BamHI, expressed from T7 promoter. A 6xHis-mKO2-HuR construction, cloned into pET30b, was expressed in Rosetta2(DE3) cells. Briefly, 1-liter of IPTG-induced (1 mM) culture was kept under shaking for 16 h at 37 °C. Cells were collected, flash-frozen in liquid nitrogen, lysed, sonicated, and clarified by centrifugation. Affinity purification performed by gravity using TALON resin (Clontech), anion exchange performed using HiTrapQ FF (GE Healthcare), size-exclusion chromatography performed using Superdex 200 10/30 prep grade (GE Healcare) – the later two performed using ÄKTA FPLC platform. DNase treated IVT, and purified 6xHis-mKO2-HuR were combined at 5 µM equimolar concentration in interaction buffer (50 mM Tris-HCl pH 7.4, 200 mM NaCl, 50% glycerol, 0.55 U/µL RNaseOUT, in DEPC-treated water). Fifty microliters of interaction buffer were deposited in CellCarrier 96-well plate (Perkin Elmer), sealed and incubated for 30 min at 25 °C. After this period, a DPSS (561 nm) laser were used to bleach and capture the

fluorescence using a PMT detector from SP8 Confocal mounted in a Leica DMi6000. The captures were performed using a PL APO CS2 63X/NA1.2 objective.

## CRISPR/Cas9 knock-in
A modified px330 plasmid to express puromycin resistance (puromycin N-acetyltransferase coding sequence removed using BamHI-XbaI from pBABE-puro and inserted in NotI-cleaved pX330; overhangs removed by Phusion DNA polymerase[120]) carrying a sgRNA targeting KGA stop codon (5′ATCTTGATGGATTGTTGTAA3′) were co-transfected with a pUC19 with constructed donor sequence (1 kb 5′ homology arm, gly-ser-thr linker, mKO2, gly-ser rich linker, 2 A self-cleaving peptide from porcine teschovirus-1, zeocin resistance, stop codon, and 1 kb 3′homology arm) into HEK293 cells. After 24 h, cells were selected with 1 µg/mL puromycin for a 24 h period. Forty-eight hours following puromycin removal, 200 µg/mL zeocin (Invitrogen) were supplemented to the medium for knock-in positive cell selection. For S-phase synchronization experiments, a double thymidine block was employed. Cells were maintained in complete DMEM medium supplemented with 100 µg/mL thymidine for 16 h, followed by 12 h thymidine-free, and final 12 h with the same concentration of thymidine. The cell cycle was evaluated by microscopy, as previously described[121]. For BT549, donor template were transferred after the puromycin resistance to a lentiGuide-puro vector which was previously cloned with the same gRNA described above. Lentivirus produced were used to transduce a BT549 cell line previously selected using lentiCas9-blast vector.

## Reporter assays
Ten thousand cells were seeded in 96-well plates. After adhering for 24 h, cells were transfected with RG6 Intron 14 or psiCHECK2 vectors using Lipofectamine 2000 (Invitrogen) following manufacturer instructions. Plates of splicing reporter assays were imaged in fluorescence microscope Operetta (Perkin Elmer) and the DsRed/GFP ratio calculated using Columbus software (Perkin-Elmer). Cells transfected with psiCHECK2 empty or carrying KGA (NM_014905) or GAC (NM_001256310) 3′UTR were evaluated using Dual-Luciferase Reporter Assay System (Promega) following manufacturer instructions.

## Glutamine dependency on oxygen consumption
Briefly, $3.0 \times 10^5$ BT549 cells were seeded in a final volume of 100 µL (complete RPMI media) in an XF24 plate (Agilent) and allowed to adhere for 1 h at 37 °C in a 5% $CO_2$ humidified incubator, when more 150 µL of same media was added and cells incubated in equal conditions for another 15 h. XFe24 sensor cartridge (Agilent) was equilibrated 1 h before the experiment, and the SeaHorse XFe24 (Agilent) was calibrated, following manufacturer instructions. Cells were washed with SeaHorse XF Assay Medium (Agilent) supplemented with 2 mM glutamine, 10 mM glucose, 1 mM sodium pyruvate, and pH adjusted to 7.4 and remained in this same medium for 1 h at 37 °C and 0% $CO_2$. Three readings were performed for the baseline oxygen consumption and more three readings after 3 µM of telaglenastat added to the medium, to evaluate the percentage of oxygen consumption irrespective of TCA anaplerosis by glutamine. Complete OCR measurements were performed using Seahorse XF Cell Mito Stress Test Kit, with 1 µM of oligomycin, 1 µM of FCCP, and 0.5 µM of Rotenone + Antimycin A, following manufacturer instructions with the modifications described above.

## Metabolic measurements
Glutamine consumption, glutamate secretion, and glutaminase activity from cell lysates were performed as previously described[8,46]. For $^{13}C/^{15}N$ labeled metabolomics, $5.10^5$ cells were plated in 100 mm cell culture dishes in RPMI with 10% dialyzed serum. After 24 h, the medium was removed, cells washed, and RPMI with 10% dialyzed serum, and 2 mM $^{13}C/^{15}N$ glutamine (Cambridge Isotope Laboratories, CNLM-1275-H-0.1) were added, including described treatments (doxycycline and or telaglenastat). Six hours later, cells were washed three times with ice-cold PBS, air dried for 1 min, quenched with 1 mL of ice-cold acetonitrile and stored at −20 °C for 10 min. Following incubation, 0.75 mL of ice-cold 53 µM Tris-HCl pH 8.0 were overlaid, and cells scraped and collected. More 1 mL of ice-cold acetonitrile and 0.75 mL of ice-cold Tris-HCl were added to wash the plate and collected in the same vial. Samples were snap-frozen in liquid nitrogen and sent to Metabolomic Center at the Department of Toxicology and Cancer Biology of the University of Kentucky for metabolomic analyses. $m + 0$ to $m + n$ indicate the different mass isotopologues for a given metabolite, where mass increases due to $^{13}C/^{15}N$-labeling[122,123].

## Migration and invasion assays
For scratch-and-wound assays, 96-well plates were treated with 0.3 mg/mL rat tail collagen type I[124] in 20 mM acetic acid for 1 h. After triple PBS wash, $3.10^4$ BT549 or $4.10^4$ MDA-MB-231 cells were plated and incubated for 16 h. Cells were washed with PBS and incubated with serum-free RPMI supplemented with 0.1% inactivated BSA for 24 h. The medium was removed, and scratches performed with the aid of a multichannel aspirator (Gilson). To avoid suspensions cells adhering to the plate, wells were washed with PBS. HEPES-buffered (25 mM, pH 7.4) DMEM high glucose (25 mM, pH 7.4) was overlaid and plates were incubated in 37 °C warm Operetta plate reader (Perkin Elmer) during the whole assay duration. Collected images were processed in Fiji[125]. For invasion assays, after the scratch procedure, cells were overlaid with 50 µL of 2 mg/mL collagen type I prepared in HEPES-buffered (25 mM, pH 7.4) DMEM high glucose with 20% FBS. Polymerization occurred at 37 °C for 30 min, culture medium was overlaid and image capture performed as previously described (protocols.io https://doi.org/10.17504/protocols.io.bgk4juyw). For Boyden chamber invasion assays, cells were incubated in 0.1% inactivated BSA for 24 h and then plated in Matrigel-covered Boyden chambers (6 µg/96-well, bottom treated with collagen type I) and incubated for 8 h at 37 °C and 5% $CO_2$ atmosphere, with 5 ng/mL of epithelial growth factor as a chemoattractant. Plates were fixed as previously described, non-invaded cells scraped using cotton swabs and DAPI-stained prior to imaging in Operetta plate reader (Perkin Elmer) and nuclei counted by Columbus.

## Statistics and reproducibility
The results are presented as the mean ± SEM as indicated and were subjected to statistical analysis using two-tailed Student's Welch-corrected $t$ test, two-way ANOVA or one-way ANOVA with the Tukey's multiple comparisons test, as appropriate. A $p$-value of <0.05 was considered statistically significant. The experiments were not randomized. The investigators were not blinded to allocation during experiments and outcome assessment. All statistical analyses were conducted using the R software, for both data analysis and visualization.

## Reporting summary
Further information on research design is available in the Nature Portfolio Reporting Summary linked to this article.

## Data availability
Source data are provided with this paper. Four Supplementary Data are provided with this paper. Public datasets used in this paper includes PRJNA30709 for RIP-Seq data of HuR, PRJNA153959 for PAR-CLIP-Seq using HEK293T, PRJNA913596 for *ELAVL1* knockout in MIA PaCa-2 cells, PRJNA140779 for *ELAVL1* knockdown and PAR-CLIP using HeLa cells, and PRJNA352155 for Breast cancer cell lines RNA-seq. Source data are provided with this paper.

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

## Acknowledgements

We thank São Paulo Research Foundation (FAPESP) for fellowships to DA (#2014/17820–3), LMR (#2014/18061-9, #2020/09535-8), and ACPM (#2016/06625-0); research grants to SMGD (#2014/15968–3, #2015/25832–4 and #2021/05726-6), and PMMMV (#2020/16030-0). We also thank LNBio for accessibility to core facilities as well as for financial support. We are very grateful to Dr. Alessandra Girasole for expert technical support. The results published here are in whole or part based upon data generated by the TCGA Research Network: https://cancergenome.nih.gov/. GAC is the Felix L. Haas Endowed Professor in Basic Science. Work in GAC's laboratory is supported by National Institutes of Health (NIH/NCATS) grant UH3TR00943-01 through the NIH Common Fund, Office of Strategic Coordination (OSC), the NCI grants 1R01 CA182905-01 and 1R01CA222007-01A1, an NIGMS 1R01GM122775-01 grant, a U54 grant #CA096297/CA096300 – UPR/MDACC Partnership for Excellence in

Cancer Research 2016 Pilot Project, a Team DOD (CA160445P1) grant, a Chronic Lymphocytic Leukemia Moonshot Flagship project, a Sister Institution Network Fund (SINF) 2017 grant, and the Estate of C. G. Johnson, Jr.

## Author contributions

S.M.G.D. and D.A. designed the study, analyzed the data, and wrote the manuscript. D.A. conducted the cell experiments, performed the bioinformatic analysis, elaborated scripts for analysis, and optimized protocols. D.A., F.C.S., and P.M.M.M.V. contributed to glutamine-dependency Seahorse experiments. A.C.P.M., I.B.N., and G.A.C. contributed to the qPCR experiment of tumor samples. L.M.R. contributed to the western blot involving a panel of breast cancer cells. All authors revised and approved the final manuscript.

## Competing interests

The authors declare no competing interests.
