## [Peer Review File · Nature Communications]

Reviewers' Comments:

Reviewer #1:

Remarks to the Author:

In this paper, authors show that HuR is involved in controlling GLS mRNA metabolism in breast cancer cells. They start providing evidences that ELAVL1 (HuR) is overexpressed in multiple tumors tissues compared to their normal counterparts by exploiting TCGA data bank. In addition, they found positive correlation between high ELAVL1 expression and both KGA and GAC mRNA levels. These are two transcripts derives from alternative splicing events of the GLS gene that encodes for glutaminase, that is a key nutrient for cancer cells, since it is involved with cell growth and tumor progression. Therefore, they try to confirm that HuR binds to GLS gene affecting its splicing mechanism. By referring to already published data of RIP-seq, RNA-seq and PAR-CLIP-seq and knocking down ELAVL1 in breast cancer cell lines they show that HuR binds to GLS intron 14 and GAC and KGA 3'UTRs, demonstrating that HuR favors the formation of KGA over GAC isoform expression and stability. Finally, authors show that HuR silencing provokes metabolic switching making the cells more dependent on glutamine consumption for proliferation and invasion, being also more susceptible to glutaminase inhibition with Telaglenastat, that is a glutaminase inhibitory compound under phase I/II in clinical trials in many therapies such as breast cancer one.

The paper has a real innovative idea as it describes the role of HuR in regulating metabolic factors with key features in driving cancer progression, that still represents a poorly explored field. This description has further implications regarding therapies, since they show that a double blockage of metabolic enzyme (glutaminase) and HuR expression leads to the decreasing of cell lines cancer traits, suggesting that combined administration of glutaminase and HuR inhibitors could have important improvements in cancer therapy.

The paper is enough well written, however it suffers of several main weaknesses. The first conceptual flaw regards the model. This paper is half a way between a pure molecular paper and therapy-related cancer one. Unfortunately, data are missing to justify both approaches. Many data are derived from meta-analyses from already existing databases. This is perfectly fine as it allows the formulation of solid hypothesis that can be verified, as the case of GLS. However, in this specific case, the authors should have performed a dedicated genome wide experiment to fully understand the role of HuR in regulating the metabolic shift described in Figure 6. This is even more evident as the rescue of GLS expression is not rescuing the HuR dependent phenotype. This is important since it has been already proposed that HuR regulates the splicing of GLS per se, and this doesn't represent an important novelty, in this work the relevant point would be the demonstration of the impact of HuR on the entire metabolic switch. The second flaw, in a therapy related cancer paper of a high impact journal, in vivo experiments are required and a specific model should be chosen.

On top of these that are some technical issues that should be solved:

-Experimental methodology and Results presentation:

Figure 2: HuR and GLS Blots in figure 2B are not clear. The GLS band in MDA-MB-157 are different from the other cell lines and the HuR blot is full of aspecific bands. Experiment in 2D is done in HEK not in breast cancer cells. The fact that the outcome of HuR silencing on GAC levels is variable along the different cell lines is reasonable but open the question of the relevance. In which type of breast cancer is this mechanism true ? This is crucial question for a cancer paper as it links the therapeutic approach to a specific gene lesion.

Figure 3: B) HuR Immunoprecipitation (Figure 3B): The enrichment of HuR in the IP samples with HuR antibody seems to be very poor, there are some doubts concerning the proper outcome given by the experiment. Furthermore, in the western blot shown in the same figure, there is discrepancy between the image and caption. HuR IgG heavy chain signal is very high in the HuR IP sample, more than HuR enrichment and the signal coming from IgG heavy chain of the IgG is completely undetectable, that is strange. For what concerns PCR in Figure 3B and 3C what authors present is not sufficient to justify HuR interaction with GLS intron 14, KGA and GAC. First of all, there is unusual unspecific background in all the samples, suggesting that the experimental procedure should be optimize in order to remove unspecific as much as possible in order to strengthen the results. Moreover, if authors' goal is to show HuR interaction with these sequences,

they should provide some quantification along with figures of the Agarose gels. In particular, they should give a graph and gel images in which signals coming from target are normalized with the one coming from a HuR non target gene. That is important to demonstrate precisely if there is a real interaction between the protein and its target. Nevertheless in order to be more quantitative and reliable, authors should check by qRT-PCR the level of the HuR bound and not-bound targets, providing some graphs regarding quantity, since images of agarose gel staining only are not acceptable anymore. 3D, FRAP experiments is not explained at all, the negative control is not described and a positive one is needed for proper quantification. A pull down experiment and a REMSA experiment would also support these data.

Figure 4: A) It is not clear how the normalization was done, meaning the control of transfection. The authors should explain. C) Why PC-3 cells? Changing model system is only adding confusion. In the text, authors jump from figure 4A to figure 5. I would suggest to rearrange. It is difficult to follow.

Figure 5) B) It is not explained why the authors did the conservation analyses. 5C) Control are missing

Figure 6) D) here authors indicate proliferation index, as well in fig7 A and fig 8A. Is this proliferation or viability ? E): Western Blot image showing HuR signals present an unspecific upper band. Please clarify

There is a usage of numerous cancer cell lines overall the paper. In Figure 5A and 2A, authors start by searching for HuR targets in already available data published in Hela Cells and they further validate their findings in breast cancer cell lines. However, it is not clear the reason why Actinomycin-D experiments in Figure 4C has been performed in PC3 cells, a human prostate cancer cell line. Since the paper is focused mainly on breast cancer and all the experiments have been performed in these cell lines, authors should provide the stability assays in at least one breast cancer cell line removing experiments performed in PC3.

Overall the manuscript, authors focus their attention on KGA and GAC mRNA expression. Since it is known that HuR regulation of the mRNA binding and stability is related also to different translational outcomes. In order to describe more deeply HuR mechanism in controlling GLS RNA metabolism, authors should provide some experiments concerning translation of the previously mentioned two mRNAs. For instance, they should perform polysome analysis in order to evaluate changes in KGA and GAC protein translation given by HuR expression modulation and put in correlation with the relative protein expression.

Reviewer #2:

Remarks to the Author:

In this manuscript "HuR controls glutaminase RNA metabolism", the author studied the role of HuR in breast cancer. The finding of HuR-controlled GLS splicing lacks novelty, as an earlier study shows that HuD has a similar function (Ref17). The author should test the role of HuD, as well as CFIm25, in their experimental system. It is worth to note that HuR is more ubiquitous than HuD. In the end, the author showed that expressing ectopic KGA after knocking down ELAVL1 did not rescue the cell growth. These data suggest KGA might be one of ELAVL1, but other targets of ELAVL1 might play a more important role in breast cancer. The conclusion of HuR-controlled glutamine metabolism is not novel, not fully supported by the data, and might be misleading. They found HuR (coded by ELAVL1) controlled GLS mRNA alternative splicing and GLS activity in multiple breast cancer cell lines. Although multiple cancer cells and shRNAs were used in this study, the data collection and presentation lacked logic. In the main figures, the authors seem to present lots of data in BT549 cells. It will be better to show all the assays in BAT549 with two different shRNAs. Similar assays in other cell lines will be helpful if the author can present the data in logical ways and discuss the different observations between the cell lines.

In the TGCA databank, ELAVL1 levels are positively correlated with KGA and GAC mRNA levels. In cell culture studies, HuR knockdown resulted in lower kidney-type glutaminase (KGA). These data seem to be controversial, and the author should further discuss this.

HuR knockdown resulted in lower kidney-type glutaminase (KGA) and higher glutaminase C (GAC),

which was not clearly stated in the abstract. They also concluded that combined chemical inhibition of GLS and silencing of ELAVL1 synergized to decrease breast cancer cell growth and invasion, which was surprising as ELAVL1 controlling GLS activity.

In 231 cells, Fig2B showed a clear single HuR band, but there are multiple bands in Fig.S5. There are also multiple bands in the other HuR blots.

Fig 2D, the altering of fluorescence signal after HuR expression and knockdown seems to be primarily driven by the different fluorescence of controls (vector and shLuc). The author should compare all those conditions in the same experiment.

It will be helpful to show the relative expression level of HuR/KGA/GAC between the selected cell lines?

Reviewer #3:

Remarks to the Author:

This is an interesting study. Here, authors show that ELAVL1 expression positively correlates with GAC and KGA. Further, authors demonstrate that HuR regulates GAC and KGA isoforms levels. Authors found that ELAVL1 KD decreased KGA with a concomitant increase in GAC. However, it lacks clarity with respect to ELAVL's metabolic regulation of glutamine metabolism. To improve the understanding related to control of glutamine metabolism by HuR, I have suggested few experiments. These experiments could enhance the conclusions authors made in this manuscript.

1. The tracing experiments may not be validated by just using one cell line. At least, three cell lines should be used. Preferably, different degree of glutamine dependency.
2. Glutamine deprivation's modulation on growth should be measured in all lines. This will establish glutamine dependency.
3. There is a huge increase in glutamine uptake when CB-839 is added. However, with and without DOX, changes are negligible in glutamine uptake. This is surprising. I am not sure why authors used 6 hours incubation with tracers. Did authors verify isotopic steady state at 6 hrs?
4. With ELAVL1 KD and/or CB839 authors notice reduced a-KG and cataplerosis. This is expected. Succinate labeling should also be shown in Fig 6.
5. What is the mechanism of ELAVL1 KD mediated reduced migration/invasion? Authors should check STAT3 , specifically serine phosphorylation.
6. Since there is a reduced TCA cycle activity when ELAVL1 is kD, how do authors explain high OCR? Authors should show the basal and time course data for OCR. They should start first in glutamine depleted media and then measure basal OCR, next add glutamine and further FCCP, antimycin. Please show time course normalized plot. It is hard to make conclusion about ELAVL's metabolic role otherwise.
7. Using GOT1 inhibitor will lead to understanding of the supplementation of the flux through these enzymes when ELAVL is KD.

POINT-BY-POINT ANSWERS TO THE REVIEWER'S COMMENTS

Reviewer #1:

General comment: *In this paper, authors show that HuR is involved in controlling GLS mRNA metabolism in breast cancer cells. They start providing evidences that ELAVL1 (HuR) is overexpressed in multiple tumors tissues compared to their normal counterparts by exploiting TGCA data bank. In addition, they found positive correlation between high ELAVL1 expression and both KGA and GAC mRNA levels. These are two transcripts derives from alternative splicing events of the GLS gene that encodes for glutaminase, that is a key nutrient for cancer cells, since it is involved with cell growth and tumor progression. Therefore, they try to confirm that HuR binds to GLS gene affecting its splicing mechanism. By referring to already published data of RIP-seq, RNA-seq and PAR-CLIP-seq and knocking down ELAVL1 in breast cancer cell lines they show that HuR binds to GLS intron 14 and GAC and KGA 3'UTRs, demonstrating that HuR favors the formation of KGA over GAC isoform expression and stability. Finally, authors show that HuR silencing provokes metabolic switching making the cells more dependent on glutamine consumption for proliferation and invasion, being also more susceptible to glutaminase inhibition with Telaglenastat, that is a glutaminase inhibitory compound under phase I/II in clinical trials in many therapies such as breast cancer one.*

The paper has a real innovative idea as it describes the role of HuR in regulating metabolic factors with key features in driving cancer progression, that still represents a poorly explored field. This description has further implications regarding therapies, since they show that a double blockage of metabolic enzyme (glutaminase) and HuR expression leads to the decreasing of cell lines cancer traits, suggesting that combined administration of glutaminase and HuR inhibitors could have important improvements in cancer therapy.

The paper is enough well written, however it suffers of several main weaknesses. The first conceptual flaw regards the model. This paper is half a way between a pure molecular paper and therapy-related cancer one. Unfortunately, data are missing to justify both approaches. Many data are derived from meta-analyses from already existing databases. This is perfectly fine as it allows the formulation of solid hypothesis that can be verified, as the case of GLS. However, in this specific case, the authors should have performed a dedicated genome wide experiment to fully understand the role of HuR in regulating the metabolic shift described in Figure 6. This is even more evident as the rescue of GLS expression is not rescuing the HuR dependent phenotype. This is important since it has been already proposed that HuR regulates the splicing of GLS per se, and this doesn't represent an important novelty, in this work the relevant point would be the demonstration of the impact of HuR on the entire metabolic switch. The second flaw, in a therapy related cancer paper of a high impact journal, in vivo experiments are required and a specific model should be chosen.

Answer: We thank the reviewer for the supportive comments and constructive criticism. We carefully read and consider each of your suggestions. Based on the reviewer's suggestions we would like to present two points:

1. To our knowledge, this is the inaugural report detailing HuR's involvement in GLS alternative splicing. We drew inspiration from Ince-Dunn et al.¹, who identified HuC/HuD (*ELAVL3/4* mouse orthologs) binding to *GLS* intron via CLIP-SEQ. HuR, encoded by *ELAVL1*, belongs to the Eukaryotic Translation Initiation Factor 3 Subunit G gene family (PantherDB: PTHR10352), presents less than 50% amino acid sequence identity compared to its paralogs HuB, HuC, and HuD (**Rebuttal figure 1**). Given the unique evolutionary path and sequence divergence of HuR, investigating its specific role in *GLS* splicing, especially in the context of breast cancer, seemed imperative.

Rebuttal figure 1 - HuR is distinct from other Hu proteins. (A) Maximum likelihood phylogenetic tree of MAFFT-aligned Hu protein family (selected from PantherDB^{2,3} Eukaryotic Translation Initiation Factor 3 Subunit G family PTHR10352) using VeryFastTree algorithm. (B) Distance (measured by identity using R-package seqinr⁴) of protein sequence between Hu proteins from mouse and human, using human RBM45 as outgroup.

2. Recognizing the absence of research on HuR's role as a metabolic regulator, we have expanded the previously presented analyses in the manuscript. We now present a pan-genomic analysis of *ELAVL1* expression using breast tumor TCGA data (**New Figure 1c-d**), utilizing the R-package singscore for signature analysis and UMAP for dimensionality reduction. This analysis revealed breast tissues with elevated *ELAVL1* expression form a distinct group compared to those with decreased levels (**New Figure 1c, left**). UMAP analysis of metabolism-related pathways further confirmed this trend (**New Figure 1c, right**). We then performed a heatmap with clustering analysis using only KEGG core metabolic pathways on breast cancer data set, which revealed a cluster (marked with blue color over the dendrogram) strongly correlated with *ELAVL1* expression (**New Figure 1d**). This cluster associates *ELAVL1* expression levels with pyrimidine, nitrogen, glyoxylate, and dicarboxylate metabolism, notably involving glutamine, indicating an enrichment compared to other clusters when we consider the presence or the absence of the metabolite inside the KEGG canonical pathways (Pathways with glutamine labelled with an "Q" in the right side). Accordingly, we have made the main text (Results: page 4, lines 124-134, Discussion: page 15, line 500).

New Supplementary Figure 01c

New Figure 1c and 1d

We examined transcriptomic data obtained with the cell line MiaPaCa5 after *ELAVL1* knockout to further complement our study. As expected, *ELAVL1* KO decreased *ELAVL1* (New Figure 2a, up) and *GLS* (New Figure 2a, down) expression levels while privileging GAC over KGA mRNA levels (Figure 2b). Notably, pathways involving nitrogen and nitrogenous bases clustered with *ELAVL1* expression, highlighting the involvement of *ELAVL1* on glutamine metabolism (page 5, lines 175-180).

New Figure 2a

Given the critical role of mitochondrial respiration in cellular metabolism, we have also expanded our SeaHorse studies (New Figure 5i-m). *ELAVL1* knock down with two different shRNAs showed an elevation in basal OCR (New Figure 5i-k), coupled with a decrease in ECAR (New Figure 5l), implying in a metabolic shift from glycolysis to aerobic respiration (New Figure 5m). This new data coupled to our previous data showing that *ELAVL1* knock

down increased glutamine uptake (Figure 5e), glutaminase activity (Figure 5f) and glutamine respiratory dependence (Figure 5h) highlights that *ELAVL1* knock down leads to an increased dependence on glutamine for mitochondrial respiration (Page 9, lines 313-319, new section “*HuR* depletion leads to a shift towards aerobic metabolism”).

New Figure 5h-m

Lastly, we used the global metabolite levels measured with GC-MS to perform a comprehensive metabolomics analysis of the BT549 cell line after *ELAVL1* knock down associated or not to telaglenastat treatment (**New Figure 6a**). While telaglenastat treatment clearly increased central carbon metabolite levels and affected mTOR signaling pathway-related metabolites (**New Figure 6a**, blue marked metabolites, and b, down), *ELAVL1* knock down decreased TCA cycle and aminoacid-related metabolites, impacting on cAMP signaling pathway-related metabolites (**New Figure 6a**, green marked metabolites, and b, up). The combination of telaglenastat treatment with *ELAVL1* knock down showed a synergistic effect (as highlighted on the PCA, Figure 6a, above, and clustering, Figure 6b, below, analyses). The new data were included in the mentioned section above (page 10, lines 323-329).

New Figure 6ab

3. Finally, regarding the issue that the manuscript would benefit from having a deeper analysis of the metabolic impact of HuR, we would like to reintroduce the breast cancer gene expression level data we had presented before as a function of *ELAVL1* expression levels (**old Suppl. 16, new Figure 1e**). While many genes of glycolysis, malate-aspartate shuttle and key lactate exporters correlate positively with *ELAVL1* expression levels, genes of fatty acid synthesis, beta-oxidation, part of the TCA cycle, and, significantly, glutamine direct uptake and metabolism, correlate with *ELAVL1* expression levels negatively. Taken together, we present evidence that HuR is a pivotal metabolic regulator in cancer; higher HuR levels impart an increased glycolytic phenotype in cells, with a decreased mitochondrial utilization of glutamine (page 5, lines 144-147).

Major point 1) *Experimental methodology and Results presentation: Figure 2: HuR and GLS Blots in figure 2B are not clear. The GLS band in MDA-MB-157 are different from the other cell lines and the HuR blot is full of aspecific bands. Experiment in 2D is done in HEK not in breast cancer cells. The fact that the outcome of HuR silencing on GAC levels is variable along the different cell lines is reasonable but open the question of the relevance. In which type of breast cancer is this mechanism true? This is crucial question for a cancer paper as it links the therapeutic approach to a specific gene lesion.*

Answer: We fully agree with the reviewer that HuR impacts on KGA and GAC protein levels differently, depending on, very likely, the genetic lesions present in the cells. As we have already discussed this issue in the manuscript and would like to bring this discussion back (page 14, lines 483-492):

“Knocking down *ELAVL1* in the cell lines revealed two different scenarios: in the cell lines where GAC and KGA were expressed in a more balanced way (two bands with similar intensities clearly seen by using an antibody that recognize both proteins; SKBR3 and MDA-MB-157), *ELAVL1* knockdown caused a shift in the isoform proportion, with a clear increase in GAC and a subsequent decrease in KGA, which reached undetectable levels; in the cell lines where GAC was already the more prominent band (BT549, MDA-MB-231, HCC38 and Hs578t), knocking down *ELAVL1* caused a visible decrease in the already faint KGA band followed by a not always apparent increase in GAC. We speculate that in this second group of cells, which are recognizably more dependent on glutamine and glutaminase for growth and survival⁵, other mechanisms cooccur to stabilize GAC levels since this isoform is the more active glutaminase^{6,7}”.

Regarding to the breast cancer genetic background in which HuR has the most clear impact on the isoforms balance levels, we would like to bring back the analysis we had previously performed where we

correlated *ELAVL1* levels with breast cancer subtypes. *ELAVL1* expression levels are higher on tumor tissues that are negative for ER, PR, TN, basal and ductal types (**Supplementary Figure 13a-f**). In these subtypes, based on our studies with cell lines and previous data from the literature, we speculate that, although high HuR levels can increase KGA levels, mechanisms other than the HuR-related ones are responsible for increasing GAC and glutamine dependence⁸. Curiously, KGA/GAC ratio is the lowest on the basal subtype (**new Suppl. Figure 13g**), the subtype that has the highest *ELAVL1* expression levels following the PAM50 classification. Overall, our data indicates that it is not HuR levels that are the decisive element behind HuR's impact on KGA/GAC balance but the overall genetic background of the breast cancer subtype. As we discussed below in the response #02 to reviewer #03, our data reveals that the critical element behind HuR direct impact on KGA/GAC levels is the cell's dependency on glutamine and glutaminase activity. Since the Luminal/HER2+ subtypes are recognizable as the less glutamine-glutaminase-dependent subtypes⁹, we would expect them to be genetic background where HuR would have the more decisive impact on KGA/GAC balance; indeed, SKBR3 cell line showed the one of the highest impact on KGA/GAC protein band upon *ELAVL1* knockdown (**Supplementary Figure 05a**). Finally, as we have also discussed below (response #02 to reviewer #03), since the TN subtype also presents variable dependency on glutamine and glutaminase¹⁰, it is possible to find situations where HuR directly impacts KGA/GAC levels; as an exemplification, the TN cell line BT549 used in the manuscript and resistant to Telaglenastat and glutamine withdraw¹⁰ was a suitable model to our studies. We included this information in the main text (page 13, lines 453-462).

g PAM50 classification

New Supplementary Figure 13g

Finally, to ensure the specificity of the HuR western blot band displayed in **Figure 2d**, we have redone the HuR western blot and updated **Figure 2d** with new data for the BT549 and MDA-MB-157 cell lines (revised blot images and separate panels in **Figure 2d**). Additionally, we have conducted a KGA knock-in experiment to express KGA with a C-terminus fusion with mKO2, in the BT549 cell line (**New figure 2g**). As expected^{11,12} and also seen for the HEK293T cell line (Supplementary Figure 5d, left), cells on the S-G2/M cell cycle phase have increased KGA mean levels.

New Figure 2g

Major point 2) Figure 3: B) HuR Immunoprecipitation (Figure 3B): The enrichment of HuR in the IP samples with HuR antibody seems to be very poor, there are some doubts concerning the proper outcome given by the experiment. Furthermore, in the western blot shown in the same figure, there is discrepancy between the image and caption. HuR IgG heavy chain signal is very high in the HuR IP sample, more than HuR enrichment and the signal coming from IgG heavy chain of the IgG is completely undetectable, that is strange. For what concerns PCR in Figure 3B and 3C what authors present is not sufficient to justify HuR interaction with GLS intron 14, KGA and GAC. First of all, there is unusual unspecific background in all the samples, suggesting that the experimental procedure should be optimized in order to remove unspecific as much as possible in order to strengthen the results.

Moreover, if authors' goal is to show HuR interaction with these sequences, they should provide some quantification along with figures of the Agarose gels. In particular, they should give a graph and gel images in which signals coming from target are normalized with the one coming from a HuR non target gene. That is important to demonstrate precisely if there is a real interaction between the protein and its target. Nevertheless in order to be more quantitative and reliable, authors should check by qRT-PCR the level of the HuR bound and not-bound targets, providing some graphs regarding quantity, since images of agarose gel staining only are not acceptable anymore. 3D, FRAP experiments is not explained at all, the negative control is not described and a positive one is needed for proper quantification. A pull down experiment and a REMSA experiment would also support these data.

Answer: We appreciate the reviewer highlighting these concerns. To clarify, the IgG control used in the IP assay shown in **Figure 3b** was sourced from a rabbit (see new information added to the Methodology section, page 18, lines 668-669), while the primary HuR antibody used in the immunoprecipitation assay was mouse-derived. Due to that, since we used an anti-mouse secondary antibody, we could detect the heavy chain on the HuR IP but not the heavy chain on the IgG IP. To bring clarity to the reader, we have now added this information to the legend of Figure 03, page 33, lines 1184-1185). To further validate our approach, we repeated the RNA IP assay using the BT549 cell line, using antibodies and IgG controls from matched species, as detailed in the **new Supplementary Figure 5h**. Additionally, we have analyzed the RIP-Assay quantitatively, using qRT-PCR. The data shows that, in BT549, HuR binds specifically to regions 2 and 4 of the KGA 3'UTR, and regions 2, 3 and 4 of the GAC's 3'UTR (**New Figure 03 d and f**). Furthermore, we now explain the FRAP experiment more thoroughly (page 7, lines 224-230). In summary, *E. coli* expressed and purified HuR-mKO2 fused protein was incubated with *in vitro* transcribed intron 14; due to the assay layout, we used, as control, a non-specific sequence transcribed from the backbone vector. Data was collected using a Leica SP8 confocal system, and show a stronger HuR binding to the GLS intron 14 sequence than to the non-specific sequence (**Figure 03i**).

New Figure 03 d-f

New Supplementary Figure 5h

Major point 3) Figure 4: A) It is not clear how the normalization was done, meaning the control of transfection. The authors should explain. C) Why PC-3 cells? Changing model system is only adding confusion. In the text, authors jump from figure 4A to figure 5. I would suggest to rearrange. It is difficult to follow.

Answer: We thank the reviewer for the raised issues. Concerning Fig 4a, we clarify that the RG6 reporter is a minigene designed to simulate exon-skipping in splicing and serves as a ratiometric sensor (**Figure 4a**). In this sense, the measured fluorescence ratio between red and green fluorescence is essential in this assay and not the total fluorescence levels. This vector includes three exons: one in box #1 (grey box numbered #1, **Figure 4a**), a skip-prone exon in box #2 (grey box numbered #2, **Figure 4a**), and a DsRed-eGFP-pA box (**Figure 4a**). We cloned the intron 14 gene sequence between boxes #1 and #2 (**Figure 4a**). According to our hypothesis, HuR promotes skipping of box #2, leading to DsRed production (511 nm emission). Conversely, *ELAVL1* knockdown maintains box #2, producing eGFP (570 nm emission). Thus, DsRed levels indicate intron 14 skipping and KGA expression, while eGFP levels indicate intron 14 maintenance and GAC expression. We only measured fluorescence in positive cells, presenting a ratio of 'red' to 'green' values. This ratiometric system, which we developed¹³ and previously used in another publication¹⁴, eliminates the need for additional transfection normalization.

Regarding Figure 4c, we have now conducted mRNA decay experiments using the BT549 cellular model. Coherently with the already seen for the overexpression in PC3 cell line, *ELAVL1* silencing reduced KGA and GAC mRNA stability (**New Figure 04d**, page 8, lines 260-266). Following the reviewer's suggestion, we have also reordered the text as to enhance readability.

New Figure 04d

Major point 4) Figure 5) B) It is not explained why the authors did the conservation analyses. 5C) Control are missing

Answer: PAR-CLIP and CLIP-seq data revealed that HuR showed multiple binding sites within *GLS'* intron 14. To perform a hypothesis-driven selection of candidate site(s) for mutation studies, we concluded that important regulatory binding sites should be conserved throughout evolution. This is now more clearly stated on page 8, lines 277-281. We selected genomes from species with GAC-specific exon retrotransposition, an apomorphic event contemporaneous to the ghost shark (*Callorhynchus milli*) speciation event to evaluate it¹⁵. Both the VISTA Browser, a tool to indicate conserved regions within vertebrate genomes, and multiple sequence alignments led to the refinement of the hundred base-sized genome location, spanning hundreds of bases to a small area of 30 bases. From within the 159-base area, we selected the proposed 8-base HuR binding site (represented in **Figure 05d**). This 8-base sequence was further mutated and shown to be involved in HuR binding (**Figure 05d**). The control used in Fig. 5d is the wild-type sequence of intron 14. This information was added to the legend of the **Figure 05d**, page 37, line 1233.

Major point 5) Figure 6) D) here authors indicate proliferation index, as well in fig7 A and fig 8A. Is this proliferation or viability? E): Western Blot image showing HuR signals present an unspecific upper band. Please clarify

Answer: We acknowledge the reviewer's concern. To clarify, in **Figures 05g, 7a, and 8a-b**, we automatically counted the number of cells (based on nuclei staining) using an automated microscope and software for cell segmentation. The number of nuclei at a specific time point was normalized against the number of seeded cells (seeding control was collected ~12 hours after seeding and thus considered time point 0h). Specifically for **Figure 05g**, we further analyzed the effect of glutamine suppression on growth by dividing the number of nuclei counted in the glutamine-suppressed condition by the number of nuclei counted in the glutamine-added condition. This methodology is now detailed in the methodology section (page 18, lines 638-640). Regarding the HuR antibody's specificity, to corroborate *ELAVL1* knockdown, presented qRT-PCR (**Figure 3** and **Supplementary Figure 6**) data shows that *ELAVL1* knockdown decreases *ELAVL1* mRNA levels in different cell lines, ranging from 60-93% (**Supplementary Figure 6**).

Major point 6) There is a usage of numerous cancer cell lines overall the paper. In Figure 5A and 2A, authors start by searching for HuR targets in already available data published in Hela Cells and they further validate their findings in breast cancer cell lines. However, it is not clear the reason why Actinomycin-D experiments in Figure 4C has been performed in PC3 cells, a human prostate cancer cell line. Since the paper is focused mainly on breast cancer and all the experiments have been performed in these cell lines, authors should provide the stability assays in at least one breast cancer cell line removing experiments performed in PC3.

Answer: We understand the reviewer's concern and have now performed actinomycin-D mRNA decay experiments using the BT549 cell line (**New Figure 04d**); the data shows that HuR impacts both KGA and GAC mRNA levels in BT549 (page 8, lines 259-261). Overall, we would like to emphasize that the use of different cell lines throughout the manuscript was done to show that the mechanism we present, although more deeply investigated on breast cancer, is of broad importance to cancer.

Major point 7) Overall the manuscript, authors focus their attention on KGA and GAC mRNA expression. Since it is known that HuR regulation of the mRNA binding and stability is related also to different translational outcomes. In order to describe more deeply HuR mechanism in controlling GLS RNA metabolism, authors should provide some experiments concerning translation of the previously mentioned two mRNAs. For instance, they should perform polysome analysis in order to evaluate changes in KGA and GAC protein translation given by HuR expression modulation and put in correlation with the relative protein expression.

Answer: We would like to reinforce that the psiCHECK reporter assay previously presented in the manuscript (**Figure 4b**) can evaluate, simultaneously, the impact of a gene's 3UTR on both mRNA stability and translation by using a reporter gene as a surrogate. Our psiCHECK data revealed that HuR has a higher impact on KGA's mRNA stability/translation, although it also affected GAC. To narrow down the protein translation information, as the reviewer suggested, we have now performed polysome enrichment experiments in BT549 after *ELAVL1* silencing (**New Figure 04e and New Supplementary Figure 07f**). *ELAVL1* knockdown decreased both KGA and GAC mRNA levels engaged in translation, as indicated by the relative polysomal to monosomic ratio (page 8, lines 261-268). Overall, this information reinforces the role of HuR in several steps of GLS' mRNA metabolism, namely splicing, mRNA stability, and mRNA translation.

New Supplementary Figure 07f

New Figure 04e

Reviewer #2:

General comment: In this manuscript "HuR controls glutaminase RNA metabolism", the author studied the role of HuR in breast cancer. The finding of HuR-controlled GLS splicing lacks novelty, as an earlier study shows that HuD has a similar function (Ref17). The author should test the role of HuD, as well as CFIm25, in their experimental system. It is worth to note that HuR is more ubiquitous than HuD.

Answer: To our knowledge, this is the first report detailing HuR involvement in GLS's alternative splicing. We drew inspiration from Ince-Dunn et al.¹, whose work used CLIP-SEQ to identify HuC/HuD (ELAVL3/4 mouse orthologs) binding to GLS intronic region leading to KGA prioritization. ELAVL1 encodes HuR and belongs to the Eukaryotic Translation Initiation Factor 3 Subunit G gene family (PantherDB: PTHR10352). HuR has less than 50% aminoacid sequence-identity compared to his paralogs HuB, HuC, and HuD (**Erro! Fonte de referência não encontrada.**). Given the HuR's unique evolutionary path and sequence divergence and its already proven involvement in several types of cancer¹⁶, in special breast cancer¹⁷⁻²², it seemed to us imperative to investigate HuR's role in GLS splicing. Also, while ELAVL1 is ubiquitously expressed in many cell types, ELAVL2, ELAVL3, and ELAVL4 are more specific to particular contexts, such as the peripheral and central neurons. Comparing the expression levels of ELAVL-Family among BRCA samples (**New Supplementary Figure 1c**), ELAVL1 is 4 log₂ units

more expressed (16 fold) than *ELAVL2*, the second family member more expressed (see also response to the general comment to reviewer #01, above).

New Supplementary Figure 01c

Finally, we found a weak but positive correlation between CFIm25 (*NUDT21*) and *ELAVL1* expression levels in TCGA-BRCA (Pearson = 0.1173), implying that in BRCA CFIm25 and HuR can co-occur and simultaneously affect KGA/GAC levels (**Rebuttal Figure 02**). The exact impact of each mechanism on *GLS* splicing and KGA/GAC choice deserves further investigation. This information was added to the discussion (page 13-14, lines 453-462).

Rebuttal Figure 02: Evaluation of the correlation between *NUDT21* and *ELAVL1* in TCGA-BRCA.

Major point 1) *In the end, the author showed that expressing ectopic KGA after knocking down ELAVL1 did not rescue the cell growth. These data suggest KGA might be one of ELAVL1, but other targets of ELAVL1 might play a more important role in breast cancer.*

Answer: We concur with the reviewer's observation and have addressed the pleiotropic effects of HuR in the discussion section of our manuscript. HuR's impact on several cellular processes, including tumorigenesis, angiogenesis, tumor inflammation, invasion, and metastasis, is well-documented and correlates with high-grade malignancy and poor prognosis in various tumors²³. This information was added to our discussion (page 13, lines 439-440). Although the role of HuR in cancer has been extensively presented in the literature, we would like to highlight that, by controlling KGA/GAC levels and potentially other genes related to glutamine metabolism, our manuscript sheds light on the unique role of HuR in controlling glutamine-dependency; importantly, we translated our findings by suggesting a potential clinical approach in which, by suppressing HuR levels, cancer cell can gain sensibility to a drug currently in clinical trials for multiple types of cancer, Telaglenastat, a GLS inhibitor.

Major point 2) *The conclusion of HuR-controlled glutamine metabolism is not novel, not fully supported by the data, and might be misleading.*

Answer: Although HuR's involvement in KGA mRNA stability was previously shown²⁴, our manuscript presents, to the best of our knowledge, the first in depth and complete evaluation on HuR's involvement in several different steps of *GLS* mRNA metabolism, including splicing, stability, and translation. Importantly, this manuscript is the first one to show HuR's involvement in *GLS* alternative splicing. We have proven HuR's binding to GAC and KGA 3'UTR mRNA by using RIP (with the data now quantified by qRT-PCR, **New Figure 3d and f**), have shown how it regulates mRNA stability (**Figure 4b-c**, **New Figure 4d**), translation (**New Figure 4e**) and splicing. Regarding splicing, we have used public data on cells manipulated to present *ELAVL1* knockdown and knockout and have presented the effect of HuR on exon differential usage and KGA choice (**Figure 02c**); we have also shown HuR direct binding to GLS's intron 14 (**Supplementary Figure 03**, **Figure 03g-h**), and functionally shown the GLS's splicing regulation by using a surrogate system (RG6 system) (**Figure 04a**, **Supplementary Figure 07**). Moreover, we thoroughly evaluated the role of HuR on metabolism and have shown that HuR drives cells to a more glycolytic phenotype (**Figure 01e**, **New Figures 05i-j**) with a decreased dependence on glutamine to mitochondrial respiration (**Figure 5h**). Specifically, we demonstrate that HuR silencing increased glutamine dependency for growth (**Figure 05g**), shifted cells to an aerobic state (**New Figure 05m**), reduced proliferation (**Figure 07a**), migration (**Figure 07b**), invasion (**Figure 07c**) of BT549, MDA-MB-231, and Hs578t. By manipulating HuR, we are the first to show the potential preferential role of GAC over KGA in glutamine dependency (page 15, lines 512-513), which deserves further investigation. Altogether, we believe that our manuscript brings significant contributions to RNA and cancer metabolism research.

New Figure 03d and f

New Figure 04d

New Figure 04e

New Figure 05h-m

Major point 3) They found *HuR* (coded by *ELAVL1*) controlled *GLS* mRNA alternative splicing and *GLS* activity in multiple breast cancer cell lines. Although multiple cancer cells and *shRNAs* were used in this study, the data collection and presentation lacked logic. In the main figures, the authors seem to present lots of data in BT549 cells. It will be better to show all the assays in BT549 with two different *shRNAs*. Similar assays in other cell lines will be helpful if the author can present the data in logical ways and discuss the different observations between the cell lines.

Answer: We appreciate the reviewer's consideration. We aimed to assess *HuR*'s role in *GLS* mRNA metabolism across various cancer backgrounds to demonstrate the mechanism's broadness. Given its already shown glutamine withdrawal- and glutaminase inhibition resistance, we have chosen BT549 to perform most of our assays as a study model. Importantly, BT549 belongs to a breast cancer subtype classification, TN, which has a great need for new therapeutic options. To get a more complete picture of BT549, we have now included additional experiments with this cell line: RIP-qRT-PCR (New Figure 03d and f), mRNA decay (New Figure 04d), and knock-in (New Figure 02g). It's important to note that experiments not featuring *shELAVL1 II* in the maintext figures were shown in the supplementary material. Please refer to supplementary Figures 05A, 06, 07a, 08, 09a, 10, and 12c, which display qRT-PCRs, splicing assays, glutamine uptake, proliferation, migration, invasion, and KGA expression rescue, respectively.

New Figure 03d and f

New Figure 04d

New Figure 02g

Major point 4) In the TCGA databank, *ELAVL1* levels are positively correlated with *KGA* and *GAC* mRNA levels. In cell culture studies, *HuR* knockdown resulted in lower kidney-type glutaminase (*KGA*). These data seem to be controversial, and the author should further discuss this.

Answer: We thank the reviewer for the observation. As demonstrated in our manuscript, *HuR* stabilizes both *GAC* and *KGA* mRNA (**Figure 04b-d**), possibly underlying the positive correlation between *ELAVL1* and *GAC* and *KGA* mRNA levels in breast tumors. Additionally, as mentioned in the Introduction (page 3, lines 77-78), several already described alternative mechanisms regulates *GLS*^{8,25-27}. *ELAVL1* expression levels are higher on tumor tissues that are negative for ER and PR, the TN, basal and ductal subtypes. In these subtypes, based on our studies with cell lines and previous data from the literature⁸, we speculate that, although high *HuR* levels can increase *KGA* levels, mechanisms other than the *HuR*-related ones are responsible for increasing *GAC* and glutamine dependence. Curiously, *KGA/GAC* ratio is the lowest on the basal subtype (**new Suppl. Figure 13g**), the subtype that has the highest *ELAVL1* expression levels following the PAM50 classification. Overall, our data indicates that it is not *HuR* levels that are the decisive element behind *HuR*'s impact on *KGA/GAC* balance but

the overall genetic background of the breast cancer subtype. As we discussed below in the response #02 to reviewer #03, our data reveals that the critical element behind HuR direct impact on KGA/GAC levels is the cell's dependency on glutamine and glutaminase activity. Since the Luminal/HER2+ subtypes are recognizable as the less glutamine-glutaminase-dependent subtypes⁹, we would expect them to be genetic background where HuR would have the more decisive impact on KGA/GAC balance; indeed, SKBR3 cell line showed the one of the highest impact on KGA/GAC protein band upon *ELAVL1* knockdown (**Supplementary Figure 05a**). Finally, as we have also discussed below (response #02 to reviewer #03), since the TN subtype also presents variable dependency on glutamine and glutaminase¹⁰, it is possible to find situations where HuR directly impacts KGA/GAC levels; as an exemplification, the TN cell line BT549 used in the manuscript and resistant to Telagelastat and glutamine withdraw¹⁰ was a suitable model to our studies. We included this information in the main text (page 13, lines 453-462).

Major point 5) *HuR knockdown resulted in lower kidney-type glutaminase (KGA) and higher glutaminase C (GAC), which was not clearly stated in the abstract*

Answer: Thank you for your observation. We altered the abstract from “*ELAVL1* knockdown in cells shifted the GAC/KGA balance” to “*ELAVL1* knockdown in cells reduced KGA and increased GAC levels” (Abstract, page 2, lines 49-50).

Major point 6) *They also concluded that combined chemical inhibition of GLS and silencing of ELAVL1 synergized to decrease breast cancer cell growth and invasion, which was surprising as ELAVL1 controlling GLS activity.*

Answer: We appreciate the reviewer highlighting this aspect. We present evidence that HuR, as a master regulator of RNA metabolism, has a pleiotropic effect on several genes and metabolite levels, causing a shift to a more glycolytic and less glutamine-dependent cellular environment (**Figure 05e-g**, **New Figure 5i-m** and **Figure 1e**). Moreover, our findings show that HuR is more directly linked to KGA levels in specific genetic backgrounds (**New Supplementary Figure 13g**). When *ELAVL1* is silenced, there is an increase in GAC levels (**Figure 03a**). Notably, GAC is often the preferred isoform in many tumor types²⁸⁻³⁰ and is known to have higher activity than KGA²⁸. GAC increased levels associated with the metabolic rewiring promoted by HuR decreased levels caused cells to depend more on glutamine, with a close association between GLS activity and TCA cycle anaplerosis. How exactly HuR causes a metabolic rewiring in cells, especially glutamine independence (apart from the GLS's isoform choice) needs further investigation.

New Figure 05i-m

g PAM50 classification

New Supplementary Figure 13g

Major point 7) In 231 cells, Fig2B showed a clear single HuR band, but there are multiple bands in Fig.S5. There are also multiple bands in the other HuR blots.

Answer: Thank you for pointing out these issues. In response, we have conducted additional western blots with a new batch of antibodies for MDA-MB-157 and BT549 (**New Figure 2b**). Importantly, we knocked down *ELAVL1*, confirmed the silencing with RT-qPCR (**Supplementary Figure 06**), and double-checked the correct HuR band with western blot (**Supplementary Figure 05**).

New Figure 02b

Major point 8) Fig 2D, the altering of fluorescence signal after HuR expression and knockdown seems to be primarily driven by the different fluorescence of controls (vector and shLuc). The author should compare all those conditions in the same experiment.

Answer: Thank you for highlighting this aspect. To clarify, the assay shown in **New Figure 02f** and **New Figure 02g** involves a CRISPR knock-in that inserts a fluorescent tag into the *GLS* gene, leading to the expression of a KGA-mKO2 fusion protein. The vectors used in this experiment (pcDNA3.1D-V5-His for ectopic HuR expression without any fluorescent tag and pLKO.puro for *ELAVL1* knockdown) do not encode any fluorescent protein. Therefore, the fluorescence observed in control cells (empty pcDNA3.1D-V5-His vector and pLKO.puro.shLuc) reflects the KGA-mKO2 fluorescence at the endogenous KGA's expression level. To compare the fluorescence levels on the control conditions and highlight the decrease in fluorescence due to *ELAVL1* knockdown and increase due to *ELAVL1* ectopic expression, we have remade the pictures applying the same intensity/contrast to equalize the controls (**New Figure 02f**).

New Figure 02f and 02g

Major point 9) It will be helpful to show the relative expression level of HuR/KGA/GAC between the selected cell lines?

Answer: In this regard, we reintroduced the previously shown HuR/KGA/GAC protein levels in several breast cancer cell lines (**Supplementary Figure 15a**), which had already been correlated to each other (**Supplementary Figure 15ab**). We also would like to reintroduce our previously shown *ELAVL1*, KGA, and GAC mRNA levels correlation analysis obtained from RNA-Seq data (**Supplementary Figure 15c**). Finally, we present, now, a barplot analysis of the RNA-seq HuR/KGA/GAC mRNA levels (**new Supplementary Figure 15d**).

New Supplementary Figure 15d

Reviewer #3:

General comment: This is an interesting study. Here, authors show that *ELAVL1* expression positively correlates with GAC and KGA. Further, authors demonstrate that HuR regulates GAC and KGA isoforms levels. Authors found that *ELAVL1* KD decreased KGA with a concomitant increase in GAC. However, it lacks clarity with respect to *ELAVL1*'s metabolic regulation of glutamine metabolism. To improve the understanding related to control of glutamine metabolism by HuR, I have suggested few experiments. These experiments could enhance the conclusions authors made in this manuscript.

Answer: We thank the reviewer for the supportive comments on our manuscript.

Major point 1) The tracing experiments may not be validated by just using one cell line. At least, three cell lines

Answer: We understand the reviewer's concern but would like to state that we performed the tracing experiment on BT549 because it was the central cell line used in the manuscript (**Figure 06**). As thoroughly discussed above, BT549 is a TN cell line resistant to glutamine withdrawal- and glutaminase inhibition, making it a very suitable model for our study. To validate our general findings on the impact of HuR on cell metabolism we used accessories RNA-seq experiments, spanning from data from breast cancer tissues and cell lines with genetic manipulation of HuR, as it will be detailed below. Moreover, we have added an additional interpretation of our metabolomic experiment using the bulk metabolomics output (traced plus non-traced metabolites). In line with that, our TCGA data dimension reduction analysis and clustering highlighted that metabolic signatures can distinguish breast cancer tissues with high *ELAVL1* expression (Figure 01c); interestingly, *ELAVL1* expression

levels correlate in the clustering with pathways related to nucleotide and nitrogen metabolism, both related to glutamine; **New Figure 01d**) and also with the expression levels of several metabolic genes that point to increased glycolysis and decreased used of glutamine in the TCA cycle (**Figure 01e**). SeaHorse analysis of BT459 after *ELAVL1* knockdown revealed a shift to a more aerobic phenotype with increased dependence on glutamine for respiration (**New Figure 05i-m**). Complementing these findings, we used a MIA PaCa-1 *ELAVL1* knock-out study to confirm that *ELAVL1* expression clusters with nitrogen and nucleotide metabolism (**New Figure 02ab**). Finally, using bulk metabolomics analysis of our metabolomic study, we revealed interesting features: 1. *ELAVL1* suppression affected the level of metabolites that concentrated in a cluster related to cAMP signaling; 2. Telaglenastat treatment affected the level of metabolites from central carbon metabolism and mTOR signaling; 3. *ELAVL1* suppression associated with Telaglenastat treatment enhanced the above-mentioned features, indicating their capacity to act in synergy (**New Figure 06ab**). In summary, we now present a more comprehensive picture of HuR's impact on cell metabolism.

New Figure 02ab

New Figure 06ab

Major point 2) *Glutamine deprivation's modulation on growth should be measured in all lines. This will establish glutamine dependency.*

Answer: We acknowledge the reviewer's suggestion and would like to state that we (and other^{31,32}) have already addressed several breast cancer cell lines regarding their glutamine dependency¹⁰. To facilitate the reviewer's analysis, we bring back our already published data, where we evaluated several cell lines regarding its Telaglenastat (CB-839)- and glutamine withdraw-sensitivity (**Rebuttal Figure 03**). As one can see, BT549 was classified as Telaglenastat- and glutamine withdraw-resistant, while MDA-MB-231 and HCC38 were classified as Telaglenastat- and glutamine withdraw-sensitive cell lines. In accordance with our study, while MDA-MB-231 and HCC38 have already increased GAC levels and were little affected by *ELAVL1* KD, BT549 presented a more pronounced shift on the KGA/GAC band intensity upon *ELAVL1* KD with a clear favoring of GAC (**Figure 2d**). SKBR3, a non-triple negative breast cancer line not reliant on *GLS*³³ confirms the HuR's critical importance in controlling KGA/GAC protein levels (**Supplementary Figure 2a**). In summary, KGA/GAC protein balance is directly linked to HuR's levels in breast cancer genetic backgrounds that dictate less dependence on glutamine and glutaminase activity, namely, the LuminalA/B and HER2+ subtypes⁹. Although the TN subtype is often described as glutaminase dependent⁹, we and other have shown that this dependence is not consistent¹⁰; in TN negative that do not rely on glutaminase, HuR exert a more pronounced effect over the KGA/GAC balance. We have now reorganized **Figure 2d** and **Supplementary Figure 05** to reflect these correlations and revised the main text to bring this information (page 6, lines 189-192).

Rebuttal Figure 03 – Glutamine dependency and glutaminase inhibition sensitivity accordingly to our other publication¹⁰.

Major point 3) *There is a huge increase in glutamine uptake when CB-839 is added. However, with and without DOX, changes are negligible in glutamine uptake. This is surprising. I am not sure why authors used 6 hours incubation with tracers. Did authors verify isotopic steady state at 6 hrs?*

Answer: Thank you for highlighting this aspect. We need to clarify that, in the tracer assay, we have only measured intracellular metabolite levels. We did not measure glutamine uptake in this assay, specifically upon

glutaminase inhibition. We have seen an increase in glutamine intracellular levels upon Telaglenastat treatment (**Figure 06c**), which is coherent with the expected glutaminase inhibition⁹. Using a different assay, though, we have shown that while *ELAVL1* KD increased glutamine uptake (**Figure 05e**, **Supplementary Figure 8a-b**) and cell lysate glutaminase activity (**Figure 05f**, **Supplementary Figure 8c**), the tracing experiment showed negligible changes in overall intracellular glutamine (**Figure 06c**) and m+6 glutamate (directly linked to glutaminase hydrolysis of glutamine) levels (**Figure 06d**). Since we have shown that *ELAVL1* KD increases glutamine use in respiration (**Figure 05h**) and decreases TCA cycle intermediates citrate, fumarate malate, and succinate (**Figure 06a**), we conclude that HuR decreased levels, by increasing the glutamine uptake, allows cells to keep up with their increased demand for this amino acid. The precise mechanism behind these observations deserves further clarification (discussion added to page 15, lines 522-523). The six-hour tracer incubation was based on the metabolomics facility *Center for Environmental and Systems Biochemistry* coordinator Teresa Fan's recommendation and is consistent with literature showing steady-state achievement after 3 hours of incubation with ¹³C Glutamine^{34,35}.

Major point 4) *With ELAVL1 KD and/or CB839 authors notice reduced α -KG and cataplerosis. This is expected. Succinate labeling should also be shown in Fig 6.*

Answer: We had chosen not to add the succinate labeling before, given the fact that the isotopologues could barely be detected in our data. Since it was required, we have added the data to the manuscript (**new Figure 6k**).

New Figure 06k.

Major point 5) *What is the mechanism of ELAVL1 KD mediated reduced migration/invasion? Authors should check STAT3, specifically serine phosphorylation.*

Answer: Acknowledging the reviewer's point, we emphasize that two well-documented pathways that link HuR to cancer cell migration and proliferation are the beta-catenin axis³⁶ and miRNA-139-3p³⁷. We conducted a western blot analysis to investigate a potential correlation between HuR levels and STAT3 Ser727 phosphorylation (**Rebuttal Figure 04**). This assay revealed no significant changes in STAT3 phosphorylation following *ELAVL1* silencing, suggesting that HuR's role in migration and proliferation may operate independently of STAT3 Ser727 phosphorylation.

BT549 tet shELAVL1

Rebuttal Figure 04: STAT3 phosphorylation in studied model.

Major point 6) *Since there is a reduced TCA cycle activity when ELAVL1 is kd, how do authors explain high OCR? Authors should show the basal and time course data for OCR. They should start first in glutamine depleted media and then measure basal OCR, next add glutamine and further FCCP, antimycin. Please show time course normalized plot. It is hard to make conclusion about ELAVL's metabolic role otherwise.*

Answer: We thank you for the observation. Our research revealed that under *ELAVL1* suppression, TCA function remains sustained due to the anaplerosis of glutamine, providing the basis for the verified synergy with Telaglenastat treatment. In our vision, the decreased TCA cycle metabolite levels registered upon *ELAVL1* knockdown are a direct result of increased utilization of these metabolites for respiration. Initially, we utilized the Agilent Seahorse XFP Mito Fuel Flex Test Kit to evaluate mitochondrial glutamine utilization (**Figure 5h**). The goal of the assay was to perform a time course OCR measurement in a glutamine-supplemented medium, followed by the blocking of the glutaminase activity (and TCA anaplerosis) using a chemical inhibitor of GLS, and then CPT1 (the fatty-acid mitochondrial transporter) with Etomoxir, and, finally, the intramitochondrial pyruvate carrier using UK5099. The results revealed that the *ELAVL1* knockdown increased the cells' reliance on glutamine for oxygen consumption. Subsequently, we verified that *ELAVL1* knockdown increased in maximum mitochondrial respiratory capacity as measured with the standard OCR assays, which use oligomycin, FCCP, and rotenone (**New Figure 05i-m**) incubation. Altogether and fully discussed above in this letter, we conclude that HuR propitiates glycolysis, with decreased dependency on glutamine use in the TCA cycle.

New Figure 05i-m

Major point 7) Using *GOT1* inhibitor will lead to understanding of the supplementation of the flux through these enzymes when *ELAVL1* is KD.

Answer: Acknowledging the reviewer's point, our TCGA data analysis shows elevated levels of *GOT1* and *GOT2* genes in BRCA samples with higher *ELAVL1* expression (**Figure 01e**). This correlation, however, is only partially mirrored in cellular models (**Rebuttal Figure 05**), where only *GOT1* levels increased upon *ELAVL1* suppression. It is important to note that *ELAVL1* silencing affected the malate-aspartate shuttle, especially evident from the reduced non-labeled aspartate levels (**Figure 06a** and **06i**). Since aspartate isotopologue mass fractions m+3 and m+4, but not m+5, decrease with *ELAVL1* knockdown (**Figure 06f**), it is a possibility that HuR impacts oxidative versus reductive glutaminolysis and cytoplasmic malate-aspartate shuttling. To point directions for future research, we've added this observation to our discussion (page 15, lines 506-508).

Rebuttal Figure 05: *GOT1* and *GOT2* levels under *ELAVL1* suppression in MiaPaCa and THP-1 cell models.

Rebuttal References:

1. Ince-Dunn, G. *et al.* Neuronal Elav-like (Hu) proteins regulate RNA splicing and abundance to control glutamate levels and neuronal excitability. *Neuron* **75**, 1067–80 (2012).
2. Mi, H. *et al.* PANTHER version 7: improved phylogenetic trees, orthologs and collaboration with the Gene Ontology Consortium. *Nucleic Acids Res* **38**, D204–D210 (2010).
3. Thomas, P. D. *et al.* PANTHER: a library of protein families and subfamilies indexed by function. *Genome Res* **13**, 2129–2141 (2003).
4. Charif, D. & Lobry, J. R. SeqinR 1.0-2: A Contributed Package to the R Project for Statistical Computing Devoted to Biological Sequences Retrieval and Analysis. in 207–232 (2007). doi:10.1007/978-3-540-35306-5_10.

5. Reis, L. M. dos *et al.* Dual inhibition of glutaminase and carnitine palmitoyltransferase decreases growth and migration of glutaminase inhibition-resistant triple-negative breast cancer cells. *Journal of Biological Chemistry* **294**, 9342–9357 (2019).
6. Ferreira, A. P. S. A. P. S. *et al.* Active glutaminase C self-assembles into a supratetrameric oligomer that can be disrupted by an allosteric inhibitor. *J Biol Chem* **288**, 28009–20 (2013).
7. Cassago, A. *et al.* Mitochondrial localization and structure-based phosphate activation mechanism of Glutaminase C with implications for cancer metabolism. *Proc Natl Acad Sci U S A* **109**, 1092–7 (2012).
8. Tambay, V., Raymond, V. A. & Bilodeau, M. Myc rules: Leading glutamine metabolism toward a distinct cancer cell phenotype. *Cancers* vol. 13 Preprint at <https://doi.org/10.3390/cancers13174484> (2021).
9. Gross, M. I., Demo, S. D. & Dennison, J. B. Antitumor Activity of the Glutaminase Inhibitor CB-839 in Triple-Negative Breast Cancer. 890–901 (2014) doi:10.1158/1535-7163.MCT-13-0870.
10. Reis, L. M. dos *et al.* Dual inhibition of glutaminase and carnitine palmitoyltransferase decreases growth and migration of glutaminase inhibition-resistant triple-negative breast cancer cells. *Journal of Biological Chemistry* **294**, 9342–9357 (2019).
11. Colombo, S. L. *et al.* Molecular basis for the differential use of glucose and glutamine in cell proliferation as revealed by synchronized HeLa cells. *Proc Natl Acad Sci U S A* **108**, 21069–74 (2011).
12. Colombo, S. L. *et al.* Anaphase-promoting complex/cyclosome-Cdh1 coordinates glycolysis and glutaminolysis with transition to S phase in human T lymphocytes. *Proceedings of the National Academy of Sciences* **107**, 18868–18873 (2010).
13. Orengo, J. P., Bundman, D. & Cooper, T. a. A bichromatic fluorescent reporter for cell-based screens of alternative splicing. *Nucleic Acids Res* **34**, e148 (2006).
14. Redis, R. S. *et al.* Allele-Specific Reprogramming of Cancer Metabolism by the Long Non-coding RNA CCAT2. *Mol Cell* **61**, 520–534 (2016).
15. Pasquali, C. C. *et al.* The origin and evolution of human glutaminases and their atypical C-terminal ankyrin repeats. *Journal of Biological Chemistry* **292**, (2017).
16. Manzoni, L. *et al.* Interfering with HuR-RNA Interaction: Design, Synthesis and Biological Characterization of Tanshinone Mimics as Novel, Effective HuR Inhibitors. *J Med Chem* acs.jmedchem.7b01176 (2018) doi:10.1021/acs.jmedchem.7b01176.
17. Xu, F. *et al.* Loss of repression of HuR translation by miR-16 may be responsible for the elevation of HuR in human breast carcinoma. *J Cell Biochem* **111**, 727–34 (2010).
18. Zhang, Z., Huang, A., Zhang, A. & Zhou, C. HuR promotes breast cancer cell proliferation and survival via binding to CDK3 mRNA. *Biomedicine & Pharmacotherapy* **91**, 788–795 (2017).

19. Heinonen, M. *et al.* Role of RNA binding protein HuR in ductal carcinoma in situ of the breast. *J Pathol* **224**, 529–539 (2011).
20. Kotta-Loizou, I., Vasilopoulos, S. N., Coutts, R. H. A. & Theocharis, S. Current Evidence and Future Perspectives on HuR and Breast Cancer Development, Prognosis, and Treatment. *Neoplasia* **18**, 674–688 (2016).
21. Licata, L. A., Hostetter, C. L., Crismale, J., Sheth, A. & Keen, J. C. The RNA-binding protein HuR regulates GATA3 mRNA stability in human breast cancer cell lines. *Breast Cancer Res Treat* **122**, 55–63 (2010).
22. Zhu, Z. *et al.* Cytoplasmic HuR expression correlates with P-gp, HER-2 positivity, and poor outcome in breast cancer. *Tumor Biology* **34**, 2299–2308 (2013).
23. Wu, M., Tong, C. W. S., Yan, W., To, K. K. W. & Cho, W. C. S. The RNA Binding Protein HuR: a Promising Drug Target for Anticancer Therapy. *Curr Cancer Drug Targets* **19**, 382–399 (2019).
24. Ibrahim, H., Lee, Y. J. & Curthoys, N. P. Renal response to metabolic acidosis: role of mRNA stabilization. *Kidney Int* **73**, 11–8 (2008).
25. Qing, G. *et al.* ATF4 Regulates MYC-Mediated Neuroblastoma Cell Death upon Glutamine Deprivation. *Cancer Cell* **22**, 631–644 (2012).
26. Wise, D. R. *et al.* Myc regulates a transcriptional program that stimulates mitochondrial glutaminolysis and leads to glutamine addiction. *Proc Natl Acad Sci U S A* **105**, 18782–7 (2008).
27. Gao, P. *et al.* c-Myc suppression of miR-23a/b enhances mitochondrial glutaminase expression and glutamine metabolism. *Nature* **458**, 762–5 (2009).
28. Cassago, A. *et al.* Mitochondrial localization and structure-based phosphate activation mechanism of Glutaminase C with implications for cancer metabolism. *Proceedings of the National Academy of Sciences* **109**, 1092–1097 (2012).
29. Ascensão, C. F. R. *et al.* N-terminal phosphorylation of glutaminase C decreases its enzymatic activity and cancer cell migration. *Biochimie* **154**, 69–76 (2018).
30. Ferreira, A. P. S. *et al.* Active Glutaminase C Self-assembles into a Supratetrameric Oligomer That Can Be Disrupted by an Allosteric Inhibitor. *Journal of Biological Chemistry* **288**, 28009–28020 (2013).
31. Quek, L. E. *et al.* Glutamine addiction promotes glucose oxidation in triple-negative breast cancer. *Oncogene* **41**, 4066–4078 (2022).
32. Gwangwa, M. V., Joubert, A. M. & Visagie, M. H. Effects of glutamine deprivation on oxidative stress and cell survival in breast cell lines. *Biol Res* **52**, 15 (2019).
33. Renna, R. K. *et al.* High-Throughput Screening Reveals New Glutaminase Inhibitor Molecules. *ACS Pharmacol Transl Sci* **4**, 1849–1866 (2021).

34. Zhang, J. *et al.* ^{13}C isotope-assisted methods for quantifying glutamine metabolism in cancer cells. in *Methods in Enzymology* vol. 542 369–389 (Academic Press Inc., 2014).
35. Woo Suk, A. & Antoniewicz, M. R. Parallel labeling experiments with $[1,2-^{13}\text{C}]$ glucose and $[U-^{13}\text{C}]$ glutamine provide new insights into CHO cell metabolism. *Metab Eng* **15**, 34–47 (2013).
36. Umar, S. M. *et al.* Quercetin Impairs HuR-Driven Progression and Migration of Triple Negative Breast Cancer (TNBC) Cells. *Nutr Cancer* **74**, 1497–1510 (2022).
37. Ni, Z.-Z. *et al.* Identification of ELAVL1 gene and miRNA-139-3p involved in the aggressiveness of NSCLC. *Eur Rev Med Pharmacol Sci* **24**, 9453–9464 (2020).

Reviewers' Comments:

Reviewer #2:

Remarks to the Author:

In this revised manuscript, the authors provided additional data to show the role of ELAVL1 (coding HuR) in glutamine metabolism. The new data partially addressed some of the previous concerns, but essential evidence is still lacking in the revised version. Related to the previous Major point 3, only one shRNA is used in all the genetic models. Furthermore, the novelty of this study remains questionable, considering the previous report of HuD in GLS. Thus, it is logical to include HuD in the essential experiments of this manuscript, which will be an important control to study the similar role of HuR in glutamine metabolism. The lack of consideration of this control dampens the quality of this study.

Reviewer #3:

Remarks to the Author:

Thanks for the modifications made based on earlier comments. I see that manuscript has been modified and reviewers tried to address the comments. However, I feel some of the experiments which were suggested were not performed. These were easy experiments and could have been done. These include including more cell lines for tracing, modified Seahorse, and GOt1 inhibitor. I am uncertain why these were not performed. Authors do provide rebuttal however, some of the experiments were critical.

Nevertheless, in light of the changes made and modifications done by authors, I am fine with recommending the acceptance of the manuscript.

POINT-BY-POINT RESPONSE TO THE REVIEWER'S COMMENTS

Reviewer #2:

General comment: *In this revised manuscript, the authors provided additional data to show the role of ELAVL1(coding HuR) in glutamine metabolism. The new data partially addressed some of the previous concerns, but essential evidence is still lacking in the revised version. Related to the previous Major point 3, only one shRNA is used in all the genetic models. Furthermore, the novelty of this study remains questionable, considering the previous report of HuD in GLS. Thus, it is logical to include HuD in the essential experiments of this manuscript, which will be an important control to study the similar role of HuR in glutaminemetabolism. The lack of consideration of this control dampers the quality of this study.*

Answer: We would like to acknowledge the reviewer for their contribution to the paper, which has been improved following the alterations performed in the first rebuttal round.

shRNA sequences used within the paper: We fully agree with the reviewer that the use of only one shRNA is insufficient to draw robust biological conclusions from silencing experiments. Therefore, starting from our initial submission, we have always presented the data regarding two shRNA sequences targeting *ELAVL1*: TRCN0000276129 (Addgene #110426, sequence CGAGCTCAGAGGTGATCAAAG, targeting the coding sequence of the mRNA) and TRCN0000276186 (Addgene #110412 & #110414, sequence TTGTTAGTGTACAACTCATTT, targeting the 3'UTR sequence of the mRNA). These sequences are referred to as sh*ELAVL1* (I) and sh*ELAVL1* (II) within the text, respectively, and are detailed in lines 624-625 of the methods section. This approach was taken to mitigate any potential region-specific bias or artifacts by targeting distinct regions of the mRNA. The data for the additional sh*ELAVL1* sequences are distributed between the main and supplementary figures. For instance, the data corresponding to Figure 2d for the other sh*ELAVL1* in different cell lines is presented in Supplementary Figure 5a; the data for Figure 3a is shown in Supplementary Figures 6a, 6b, and 6c; and so forth.

HuD level in breast cancer: Our study is the first one to detail HuR's involvement in the alternative splicing of *GLS*. We were inspired by the work of Ince-Dunn et al.¹, who identified the binding of HuC/HuD (the mouse orthologs of *ELAVL3/4*) to *GLS* intron via CLIP-SEQ. HuR, encoded by the *ELAVL1* gene, is a member of the Eukaryotic Translation Initiation Factor 3 Subunit G gene family (PantherDB: PTHR10352). It shares less than 50% amino acid sequence identity with its paralogs HuB, HuC, and HuD. Given HuR's distinct evolutionary trajectory and sequence divergence from HuD, we wanted to investigate its specific role in the splicing of *GLS*, particularly in the context of breast cancer.

To evaluate the expression levels of *ELAVL*-family genes in breast cancer, we analyzed the BRCA-TCGA database, which offers the sequencing depth necessary to detect even the lowest expressed transcripts in the genome. As shown in **Supplementary Figure 1c** (which we reproduced bellow), in breast cancer, *ELAVL4* (which codify HuD) expression level is not significantly altered in tumor samples compared to non-paired normal samples. Specifically, the median of the length-

normalized FPKM expression level is 360 times higher for *ELAVL1* compared to *ELAVL4* in the tumor tissue data (modified **Supplementary Figure 01c**, also included below for your convenience).

We further assessed the expression of *ELAVL4* in 19 breast cancer cell lines using datasets obtained from databanks²⁻⁴. These data were already explored in the Supplementary Figure 14. For all evaluated cell lines, *ELAVL1* exhibited the higher Expectation-Maximization (RSEM) values compared to *ELAVL2*, *ELAVL3* and *ELAVL4*. In BT549, HCC1599, HCC1187, HCC70, HCC1143, MDA-MB-468, and MCF10A, only *ELAVL1* and *ELAVL2* were detected. In 12 out of the 19 cell lines, *ELAVL2*, *ELAVL3* and *ELAVL4* were not detected at all (**Rebuttal Figure 1**).

replicate from independent laboratories, deposited under the Sequence Read Archive (SRA) and described in the paper.

We then employed qPCR to evaluate whether *ELAVL4* was expressed when *ELAVL1* was knocked down. *ELAVL4* levels were quantified based on a linear regression obtained with the amplification of a synthesized amplicon (Rebuttal Figure 2a). In all tested cell lines (BT549, MDA-MB-231, and Hs578t), in both control and *ELAVL1* knockdown conditions induced by the shRNAs sequences presented in the manuscript, the number of copies quantified (25×10^5) fell outside the linear quantification range of the standard curve (Rebuttal Figure 2b). These data indicates that *ELAVL4* expression is not detectable in a quantifiable manner in these cell lines, in our working conditions. Importantly, *ELAVL4* levels did not increase when *ELAVL1* was knocked down as revealed by both absolute (Rebuttal Figure 2b) and relative quantification (Rebuttal Figure 2c). All together, these data reveal that *ELAVL4* has a very low expression level in breast cancer, especially when compared to *ELAVL1*, indicating that *ELAVL4* does not play a role in regulating *GLS* isoforms levels in breast cancer.

Rebuttal Figure 2 – Quantification of *ELAVL4* Expression Using qPCR. (a) Synthetic target amplification under serial dilution to evaluate the limit of detection of the qPCR reaction. (b) Absolute quantification of *ELAVL4* in various cell lines under *ELAVL1* silencing; the quantified expression levels fall outside the linear range of qPCR, and indicating that the expression levels are close to the detection limit. (c) Relative quantification of *ELAVL4* expression in the same cell lines, normalized to 18S rRNA as a reference gene. Error bars represent the standard error of the mean (s.e.m.).

Reviewer #3:

General comment: Thanks for the modifications made based on earlier comments. I see that manuscript has been modified and reviewers tried to address the comments. However, I feel some of the experiments which were suggested were not performed.

These were easy experiments and could have been done. These include including more cell lines for tracing, modified Seahorse, and G0t1 inhibitor. I am uncertain why these were not performed. Authors do provide rebuttal however, some of the experiments were critical. Nevertheless, in light of the changes made and modifications done by authors, I am fine with recommending the acceptance of the manuscript.

Answer: We would like to thank the reviewer for their time and effort in providing a fair and dedicated evaluation of our manuscript. We are grateful for your constructive feedback and are pleased that you find the manuscript acceptable for publication with the modifications made. We believe that the changes implemented have enhanced the quality of our work and your suggestions were fundamental to make it more suitable for publication in Nature Communications.

Reviewers' Comments:

Reviewer #2:

Remarks to the Author:

The authors addressed my questions about HuD. They provided more evidence showing the big difference between HuD and HuR, and the relative lower ELAVL4 expression.

They also include data with two different shRNA in multiple cells and assays. However, the current figure layout mainly shows data with one shRNA in the main figures. It is not easy for readers to chase the missing data of the second shRNA in the supplementary. Additionally, most assays should be done with both shRNA within the same experiment. Thus, it is very common to include both shRNA in the main figures, at least for key experiments.

ANSWER TO THE REVIEWER'S COMMENT

Reviewer #2:

General comment: *The authors addressed my questions about HuD. They provided more evidence showing the big difference between HuD and HuR, and the relative lower ELAVL4 expression. They also include data with two different shRNA in multiple cells and assays. However, the current figure layout mainly shows data with one shRNA in the main figures. It is not easy for readers to chase the missing data of the second shRNA in the supplementary. Additionally, most assays should be done with both shRNA within the same experiment. Thus, it is very common to include both shRNA in the main figures, at least for key experiments.*

Answer: We thank the reviewer for the supportive comments on the *ELAVL1*, *ELAVL2*, *ELAVL3*, and *ELAVL4* data, and for the constructive criticism regarding our figure assembly and layout. In response to the feedback, we have made the following adjustments to the figures:

- 1) Western Blots: Previously located in Supplementary Figure 05a, the western blots for the inducible system using sh*ELAVL1* and sh*ELAVL1* II in both BT549 and MDA-MB-231 models have been moved to Figure 02d.
- 2) qPCR Data: The constitutive BT549's shRNA qPCR data from Figure 03a have been exchanged with data from Supplementary Figure 06a, which includes a doxycycline-inducible system using both shRNAs.
- 3) Glutamine Uptake: Data on glutamine uptake in BT549, originally in Figure 05e, have been relocated to Supplementary Figure 08a. Conversely, data involving BT549, MDA-MB-231, and Hs578t sh*ELAVL1* II from Supplementary 08a have been moved to Figure 05e.
- 4) Proliferation and Wound Healing Data:
Proliferation data from sh*ELAVL1* II for BT549, MDA-MB-231, and Hs578t, previously in Supplementary Figure 09a, are now included in Figure 07.
Wound healing data for sh*ELAVL1* II from BT549 and MDA-MB-231, originally in Supplementary Figure 10a, have been moved to the main text and figures.
- 5) Boyden Chamber Invasion Data: The invasion data for sh*ELAVL1* II involving the cell lines BT549, MDA-MB-231, and Hs578t, previously part of Supplementary Figure 10c, have been integrated into the main figures.
- 6) Proliferation data with Telaglenastat treatment:
Data presented in Figure 08a, middle, were included with the sh*ELAVL1* II results for the telaglenastat combination.

All modifications have been carefully implemented in the figures, with respective captions and mentions in the main text updated to reflect these changes. We believe these adjustments enhance the clarity and accessibility of the data as per the reviewer's suggestions.

Reviewers' Comments:

Reviewer #2:

Remarks to the Author:

Glad to see the rearranging of the figures, showing most of the data for 2 shRNA side-by-side.